# Uncharacterized yeast gene *YBR238C*, an effector of TORC1 signaling in a mitochondrial feedback loop, accelerates cellular aging via *HAP4-* and *RMD9-*dependent mechanisms

Mohammad Alfatah[1]*, Jolyn Jia Jia Lim[1], Yizhong Zhang[1], Arshia Naaz[2], Trishia Yi Ning Cheng[1], Sonia Yogasundaram[1], Nashrul Afiq Faidzinn[1], Jovian Jing Lin[1], Birgit Eisenhaber[1,2,3], Frank Eisenhaber[1,2,3,4]

[1]Bioinformatics Institute (BII), Agency for Science, Technology and Research (A*STAR), Singapore, Singapore; [2]Genome Institute of Singapore (GIS), Agency for Science, Technology and Research (A*STAR), Singapore, Singapore; [3]LASA – Lausitz Advanced Scientific Applications gGmbH, Weißwasser, Germany; [4]School of Biological Sciences (SBS), Nanyang Technological University (NTU), Singapore, Singapore

*For correspondence:
alfatahm@bii.a-star.edu.sg

Competing interest: The authors declare that no competing interests exist.

**Abstract** Uncovering the regulators of cellular aging will unravel the complexity of aging biology and identify potential therapeutic interventions to delay the onset and progress of chronic, aging-related diseases. In this work, we systematically compared genesets involved in regulating the lifespan of *Saccharomyces cerevisiae* (a powerful model organism to study the cellular aging of humans) and those with expression changes under rapamycin treatment. Among the functionally uncharacterized genes in the overlap set, *YBR238C* stood out as the only one downregulated by rapamycin and with an increased chronological and replicative lifespan upon deletion. We show that *YBR238C* and its paralog *RMD9* oppositely affect mitochondria and aging. *YBR238C* deletion increases the cellular lifespan by enhancing mitochondrial function. Its overexpression accelerates cellular aging via mitochondrial dysfunction. We find that the phenotypic effect of *YBR238C* is largely explained by *HAP4-* and *RMD9-*dependent mechanisms. Furthermore, we find that genetic- or chemical-based induction of mitochondrial dysfunction increases TORC1 (Target of Rapamycin Complex 1) activity that, subsequently, accelerates cellular aging. Notably, TORC1 inhibition by rapamycin (or deletion of *YBR238C*) improves the shortened lifespan under these mitochondrial dysfunction conditions in yeast and human cells. The growth of mutant cells (a proxy of TORC1 activity) with enhanced mitochondrial function is sensitive to rapamycin whereas the growth of defective mitochondrial mutants is largely resistant to rapamycin compared to wild type. Our findings demonstrate a feedback loop between TORC1 and mitochondria (the TORC1–MItochondria–TORC1 (TOMITO) signaling process) that regulates cellular aging processes. Hereby, *YBR238C* is an effector of TORC1 modulating mitochondrial function.

## eLife assessment

This **valuable** study identifies an uncharacterized yeast gene regulating chronological lifespan in a mitochondrial-dependent pathway. The approach to identify and characterise this new gene is appealing, but the evidence in support of some of the major conclusions is **incomplete**. The paper

focuses on chronological lifespan and mitochondrial function, and it will be of interest to yeast biologists working in metabolism and aging.

## Introduction

Healthy aging is crucially determined by cellular functions, and their defective status is associated with premature dysfunction and/or depletion of critical cell populations and various aging-associated pathologies such as neurodegenerative diseases, cancer, cardiovascular disorders, diabetes, sarcopenia, and maculopathy (*Kennedy et al., 2014*; *Franceschi et al., 2018*; *Tchkonia and Kirkland, 2018*; *Gladyshev, 2016*; *Richardson, 2021*). Advancements in biomedical research including genome-wide screening in different aging model organisms, identified several biological pathways that support the unprecedented progress in understanding overlapping profiles between aged cells and different chronic aging-associated diseases (*Cohen et al., 2022*; *McCormick et al., 2015*; *Powers et al., 2006*; *Hamilton et al., 2005*). In principle, uncovering the molecular mechanisms that drive cellular aging identifies potential drug targets and can fuel the development of therapeutics to delay aging and increase healthspan (*López-Otín et al., 2013*; *López-Otín et al., 2023*; *Partridge et al., 2020*; *Singh et al., 2019*).

Given the evolutionary conservation of many aging-related pathways, yeast is one of the aging model organisms that have been extensively used to study the biology of human cellular aging by analyzing chronological lifespan (CLS) and replicative lifespan (RLS) under various conditions (*Longo and Fabrizio, 2015*; *Longo et al., 2012*; *Fontana et al., 2010*; *Zimmermann et al., 2018a*). The CLS is the duration of time that a non-dividing cell is viable. This is a cellular aging model for post-mitotic human cells such as neurons and muscle cells. The RLS defines as the number of times a mother cell divides to form daughter cells. Such experiments provide a replicative human aging model for mitotic cells such as stem cells. Genome-wide or individual gene deletion strains' screening identified thousands of genes affecting cellular lifespan. These lists are a rich resource to identify unique genetic regulators, functional networks, and interactions of aging hallmarks relevant to cellular lifespan (*Kennedy et al., 2014*; *Gladyshev, 2016*; *Cohen et al., 2022*; *López-Otín et al., 2013*; *López-Otín et al., 2023*; *Partridge et al., 2020*; *Singh et al., 2019*). At the same, these lists are rich in genes of unknown function, a class of genes that, unfortunately, got increasingly ignored by the attention of research teams (*Eisenhaber, 2012*; *Sinha et al., 2018*; *Tantoso et al., 2023*).

We systematically examined the available genetic data on aging and lifespan for budding yeast, *Saccharomyces cerevisiae*, from various sources. Our bioinformatics analyses revealed consensus lists of genes regulating yeast CLS and RLS. These gene sets were compared with lists of genes the expression of which changed under Target of Rapamycin Complex 1 (TORC1) inhibition with rapamycin. When we turned our attention to the functionally uncharacterized genes involved, we found *YBR238C* the only one among the latter that increases both CLS and RLS upon deletion and that is downregulated by rapamycin. Transcriptomics and biochemical experiments revealed that *YBR238C* negatively regulates mitochondrial function, largely via *HAP4-* and *RMD9*-dependent mechanisms, and thereby affects cellular lifespan. Surprisingly, *YBR238C* and its paralog *RMD9* oppositely influence mitochondrial function and cellular aging.

Our chemical genetics and metabolic analyses unravel a feedback loop of the interaction of TORC1 with mitochondria that affect cellular aging. *YBR238C* is an effector of TORC1 that modulates mitochondrial function. We also show that mitochondrial dysfunction induces TORC1 activity enhancing cellular aging. In turn, TORC1 inhibition in yeast and human cells with mitochondrial dysfunction enhances their cellular survival.

## Results

### Genome-wide survey of genes affecting cellular lifespan in yeast in accordance with literature and public databases

Lists of scientific literature instances mentioning a yeast gene as affecting lifespan were downloaded from the databases SGD (*Saccharomyces cerevisiae* genome database) (*Engel et al., 2022*) and GenAge (*Townes et al., 2020*). In most cases, the experiments referred to are gene deletion

**Table 1.** Numbers of known aging-associated genes (AAGs) reported in the scientific literature to affect (increase/decrease chronological lifespan [CLS]/replicative lifespan [RLS]) under a variety of conditions.

The gene lists were extracted, after processing, from downloaded files (as of November 8, 2022) using the databases SGD (*Engel et al., 2022*) and GenAge (*Townes et al., 2020*). The actual gene lists are available in *Figure 1—source data 1*. The signs '+' and '−' indicate that the respective annotation property is present or missing in that group of genes. We present the total number of genes in each category as the number of, as a trend, severely understudied/uncharacterized genes among those. The columns 'RUG' and 'RDG' show the numbers of rapamycin-up- and -downregulated genes in each of the 15 categories.

| Category | CLS increase | CLS decrease | RLS increase | RLS decrease | Number of genes | Understudied genes | RUG | RDG |
|---|---|---|---|---|---|---|---|---|
| 1 | + | − | − | − | 328 | 20 | 68 | 62 |
| 2 | − | + | − | − | 615 | 22 | 108 | 73 |
| 3 | − | − | + | − | 349 | 19 | 73 | 64 |
| 4 | − | − | − | + | 318 | 15 | 43 | 56 |
| 5 | + | + | − | − | 108 | 0 | 17 | 18 |
| 6 | + | − | + | − | 72 | 2 | 12 | 26 |
| 7 | + | − | − | + | 72 | 3 | 9 | 18 |
| 8 | − | + | + | − | 120 | 3 | 14 | 28 |
| 9 | − | + | − | + | 180 | 0 | 24 | 34 |
| 10 | − | − | + | + | 59 | 0 | 8 | 14 |
| 11 | + | + | + | − | 29 | 1 | 7 | 7 |
| 12 | + | + | − | + | 45 | 1 | 7 | 9 |
| 13 | + | − | + | + | 30 | 1 | 2 | 11 |
| 14 | − | + | + | + | 55 | 1 | 5 | 11 |
| 15 | + | + | + | + | 19 | 0 | 0 | 2 |

phenotype studies. After processing the files for the mentioning of 'increase/decrease' of 'chronological lifespan' (CLS) and/or 'replicative lifespan' (RLS) as well as the suppression of gene duplicates, we found 2399 entries with distinct yeast genes in 15 categories reported to increase/decrease CLS and/or RLS under various conditions (*Table 1*). We collectively call them aging-associated genes (AAGs). Notably, about one-third of the total yeast genome belongs to that category. Downloaded files (as of November 8, 2022), description of the processing details, and all the resulting gene lists are available in *Figure 1—source data 1*.

Whereas most of the genes (1610, 67%) have been mentioned for just one of the four scenarios ('CLS increase', 'CLS decrease', 'RLS increase', and 'RLS decrease'), the remaining genes have been described for several alternative and/or even opposite/conflicting outcomes. Nineteen genes are brought up in context with all four scenarios. As the experimental conditions varied among the reports and the gene networks are complex, this is not necessarily a contradiction. Yet (see below) some of the cases are actually annotation errors at the database level.

We explored the potential enrichment of gene ontology (GO) terms among the genes involved in the 15 categories with the help of DAVID WWW server (*Sherman et al., 2022*). The most significant result is the enrichment of ontology terms related to mitochondrial function among the genes annotated with the single qualifier 'CLS decrease'. The top mitochondrial term cluster has an enrichment score 18.2 (gene-wise p values and Benjamini–Hochberg values all below 6.e−13); a second one related to mitochondrial translation has the score 4.7. Enrichment of mitochondrial ontology terms has also been observed for 'CLS decrease' in combination with 'CLS increase' or 'RLS increase/decrease'.

Every other signal from the ontology is much weaker. Term enrichment (with scores near 2 or better) related to autophagy and vesicular transport pop up for CLS-annotated genes whereas terms connected with translation, DNA repairs, telomeres, protein degradation, and signaling are observed with RLS-tagged gene lists.

We also explored the presence of uncharacterized or severely under-characterized genes in the 15 categories of AAGs (*Table 1*). In total, 944 genes are annotated in SGD as coding for a protein of unknown function. Additionally, we considered genes without dedicated gene name (only with six-letter locus tag) as dramatically under-characterized. Comparison of this combined list with those in the 15 categories reveals 88 severely understudied AAGs that are candidates for enhanced attention from the scientific community. Thus, functionally insufficiently characterized genes have a great role in cellular aging-related processes.

## Rapamycin response genes overlap with the AAGs

Nutrient sensing dysregulation is one of the aging hallmarks (*López-Otín et al., 2013*; *López-Otín et al., 2023*). The conserved protein complex TORC1 senses nutrients such as amino acids and glucose and links metabolism with cellular growth and proliferation (*Loewith and Hall, 2011*; *Saxton and Sabatini, 2017*; *Liu and Sabatini, 2020*; *González and Hall, 2017*). TORC1 positively regulates aging, and its inhibition increases lifespan in various eukaryotic organisms including yeast and mammals (*Partridge et al., 2020*; *Saxton and Sabatini, 2017*; *Liu and Sabatini, 2020*; *Bitto et al., 2016*; *Johnson et al., 2013a*). The drug rapamycin, initially discovered as an antifungal natural product produced by *Streptomyces hygroscopicus*, inhibits TORC1 and increases lifespan (*Figure 1—figure supplement 1A, B*; *Vézina et al., 1975*; *Selvarani et al., 2021*).

To explore the connection between nutrient signaling and cellular aging we mapped the TORC1-regulated genes with AAGs. We first identified rapamycin response genes (RRGs) by transcriptomics analysis (RNA sequencing [RNA-Seq] for yeast *S. cerevisiae* BY4743 cells treated with rapamycin and dimethyl sulfoxide [DMSO] control; *Figure 1—figure supplement 1C*). Relevant measurement results, lists of 2365 RRGs and supplementary methodical comments are available in *Figure 1—source data 2*.

As overlap of two RNA-Seq data analysis methods (see Methods), we identified 1155 rapamycin upregulate genes (RUG) and 1210 rapamycin downregulated genes (RDG) (*Figure 1—figure supplement 1D-F*; *Figure 1—source data 2*). RNA-Seq results were confirmed for some genes by quantitative reverse transcription-polymerase chain reaction (qRT-PCR; *Figure 1—figure supplement 2A*). We also checked the differential gene expressions in prototrophic yeast *S. cerevisiae* strain CEN.PK and found similar results as with the BY4743 strain (*Figure 1—figure supplement 2B*). To note, our transcriptomics analyses are, as a trend, consistent with previous RNA-Seq studies carried out under partially different experimental conditions (*Gowans et al., 2018*; *Alfatah et al., 2021*).

We mapped the RRGs (RUG and RDG) with AAGs. Among the 2399 AAGs names, 397 and 433 (in total 830) re-occur in the lists of RUG and RDG, respectively. Thus, the overlap with AAGs is nearly 35%. That is, TORC1 controls about one-third of the AAGs. *Table 1* shows how many AAGs are up- and downregulated by rapamycin treatment for each of the 15 categories with regard to CLS/RLS increase/decrease. Notably, the order of magnitude for the respective numbers of RUG and RDG is the same for all 15 studied subgroups.

Since rapamycin increases the lifespan by an inhibitory effect on TORC1 activity, it appears most interesting to focus on AAGs with a deletion phenotype of increased CLS/RLS and being downregulated by rapamycin application. A manual analysis of these gene lists reveals that, among the uncharacterized or severely understudied AAGs, there is a single one known to increase both CLS and RLS upon deletion and being downregulated by rapamycin treatment. This gene is *YBR238C*.

## Uncharacterized genes mapping unravels the role of *YBR238C* in cellular aging

Not much is known about *YBR238C* besides its effect on lifespan (to increase CLS and RLS), its mitochondrial localization (*Huh et al., 2003*), its transcriptional upregulation by TORC1 and the existence of the paralog *RMD9*. The encoded protein has 731 amino acid residues. Sequence architecture analysis with the ANNOTATOR (*Eisenhaber et al., 2016*) reveals an intrinsically unstructured region over the first ca. 130 residues (first, a long polar but uncharged run followed by a histidine/asparagine-rich

region beginning with position 83) and a pentatricopeptide repeat region (residues 130–675, e.g., due to a HHpred *Zimmermann et al., 2018b* hit to structure 7A9X chain A with *E*-value 2.e−56). Given the sequence homology, we hypothesize that the protein encoded by *YBR238C* is involved in RNA binding as its paralog *RMD9* with a similar globular segment (*Hillen et al., 2021*).

To note, *YBR238C* carries the conflicting annotations in SGD for increased and decreased RLS upon deletion. Unfortunately, there is not a single direct report about *YBR238C* listed in the scientific literature at the time of writing. There are a few genome-wide deletion strain studies that identified *YBR238C* as one of the gene that increases CLS (*Burtner et al., 2011*) and RLS (*McCormick et al., 2015*; *Delaney et al., 2011*; *Kaeberlein et al., 2005*; *Schleit et al., 2013*). However, the SGD database wrongly documented one of the latter studies as evidence for a decreased RLS phenotype of *YBR238C* (*Schleit et al., 2013*). Thus, this examination allows us to claim *YBR238C* as the only uncharacterized RDG causing CLS and RLS increase upon deletion.

First, we confirmed that *YBR238C* is indeed an RRG by qRT-PCR expression analysis in both yeast backgrounds BY4743 and CEN.PK (*Figure 1A, B*). Given that only a single genome-wide deletion strain study identified *YBR238C* as a gene that enhances CLS (*Burtner et al., 2011*), we further tested the role of *YBR238C* in CLS of yeast. CLS was analyzed in BY4743 and CEN.PK strains using three different outgrowth survival methods (see detail in methods section). Cell survival of wild type and *ybr238cΔ* strains was analyzed at various age time points. We found higher CLS of *ybr238cΔ* cells compared to wild-type cells (*Figure 1C–F*). Together, these results confirmed that rapamycin inhibits the expression of *YBR238C*, and deletion of this gene indeed increases the cellular lifespan.

## Transcriptomics analysis reveals the longevity gene expression signatures of *ybr238cΔ*mutants

The transcriptome of the long-lived *ybr238cΔ* mutant was compared with that of the wild type (*Figure 2—figure supplement 1A*). Applying standard significance criteria, we found 326 genes up- and 61 genes downregulated in the*ybr238cΔ* mutant compared to wild type (*Figure 2—figure supplement 1B*, *Figure 2—source data 1*). Thus, the transcriptome of the *ybr238cΔ* mutant is very distinct from that of the wild type.

Notably, we see several major metabolic changes. For the *ybr238cΔ* mutant, genes in mitochondrial metabolic processes such as oxidative phosphorylation and aerobic respiration show predominant enrichment (*Figure 2A, B*,*Figure 2—source data 1*). Since *YBR238C* expression is regulated by TORC1 (*Figure 1A, B*), it is not surprising that we observe similarities in the profiles of upregulated differentially expressed genes (DEGs) for the *ybr238cΔ* mutant and for the case of treating the wild type with rapamycin (*Figure 2—figure supplement 1C, D*).

Next, we experimentally tested whether the transcriptome longevity signatures are associated with enhanced mitochondrial metabolism, whether the cellular energy level has gone up and cellular stress responses are induced with a switch to oxidative metabolism (*Petti et al., 2011*; *Galluzzi et al., 2012*). Indeed, our metabolic analysis revealed an increased ATP level in *ybr238cΔ* mutants compared to wild-type cells (*Figure 2C, D*, *Figure 2—figure supplement 1E*). ATP generation through the oxidative phosphorylation (OXPHOS) system is regulated by mitochondrial DNA (mtDNA) (*Shadel, 1999*). We observed a higher mtDNA copy number in the *ybr238cΔ* mutant compared to wild-type cells (*Figure 2—figure supplement 1F*). Hence, the *ybr238cΔ* mutation rewires the cellular metabolism to promote resource-saving ways of energy production as the upregulated expression of OXPHOS machinery subunits truly boosts ATP synthesis in *ybr238cΔ* mutant cells.

The enhanced mitochondrial function in *ybr238cΔ* mutants does also improve the protection against reactive oxygen species (ROS). Among the 150 transcription factors (TFs) that control the upregulated DEG of the *ybr238cΔ* mutant (*Figure 2—source data 1*), 13 TFs are significantly overrepresented within the upregulated DEGs (*Figure 2—source data 1*). Importantly, we found that the stress response controlling TF *MSN4* is upregulated in *ybr238cΔ* mutants (*Figure 2E, F*, *Figure 2—figure supplement 2A, B*; *Longo et al., 2012*; *Lin et al., 2002*). Concomitantly, we found less ROS level in *ybr238cΔ* mutant compared to wild-type cells (*Figure 2G, H*, *Figure 2—figure supplement 1G, H*) and is resistant to $H_2O_2$-induced oxidative stress toxicity (*Figure 2—figure supplement 1I*).

Notably, the TF-regulated activation of stress response pathways (thioredoxins, molecular chaperones, etc.) (*Turcotte et al., 2010*; *Schüller, 2003*) and the switch from fermentation to respiration are associated with delayed cellular aging (*Petti et al., 2011*; *Galluzzi et al., 2012*; *Jensen and Jasper,*

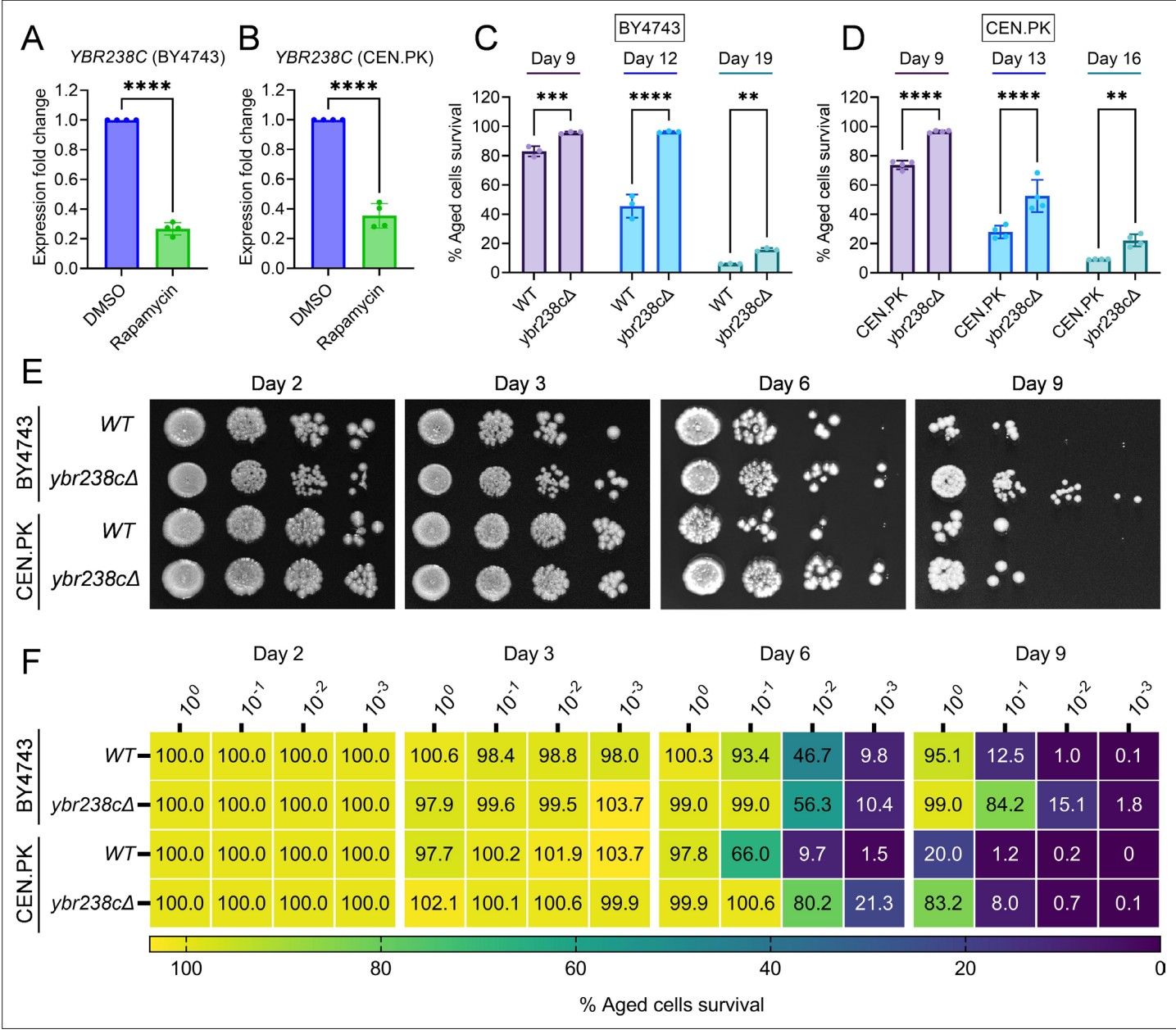

**Figure 1.** *YBR238C* deletion increases the cellular lifespan. (**A and B**) Expression analysis of *YBR238C* gene by quantitative reverse transcription-polymerase chain reaction (qRT-PCR) in yeast *Saccharomyces cerevisiae* genetic backgrounds BY4743 (**A**) and CEN.PK113-7D (**B**). qRT-PCR was performed using RNA extracted from logarithmic-phase cultures grown in synthetic defined medium supplemented with auxotrophic amino acids for BY4743 (**A**) and only the synthetic defined medium for CEN.PK113-7D (**B**). Data are represented as means ± standard deviation (SD) (*n* = 4). ****p < 0.0001 based on two-sided Student's *t*-test. (**C and D**) Chronological lifespan (CLS) of the wild type and *ybr238cΔ* mutant was assessed in synthetic defined medium supplemented with auxotrophic amino acids for BY4743 (**C**) and only the synthetic defined medium for CEN.PK113-7D (**D**) strains in 96-well plate. Aged cells survival was measured relative to the outgrowth of day 2. Data are represented as means ± SD (*n* = 3) (**C**) and (*n* = 4) (**D**). **p < 0.01, ***p < 0.001, and ****p < 0.0001 based on two-way analysis of variance (ANOVA) followed by Šídák's multiple comparisons test. (**E, F**) CLS of the wild type and *ybr238cΔ* mutant was performed in synthetic defined medium supplemented with auxotrophic amino acids for BY4743 and only the synthetic defined medium for CEN.PK113-7D strains using flasks. Outgrowth was performed for 10-fold serial diluted aged cells onto the YPD agar plate (**E**) and YPD medium in the 96-well plate (**F**). The serial outgrowth of aged cells on agar medium was imaged (**E**) and quantified the survival relative to outgrowth of day 2 (**F**). A representative of two experiments for (**E**) and (**F**) is shown.

The online version of this article includes the following source data and figure supplement(s) for figure 1:

**Source data 1.** Genome-wide survey of aging-associated genes (AAGs) in *Saccharomyces cerevisiae*.

**Source data 2.** Rapamycin response genes and aging-associated genes mapping.

*Figure 1 continued on next page*

Figure 1 continued

**Figure supplement 1.** Transcriptomics analysis of rapamycin to identify the Target of Rapamycin Complex 1 (TORC1)-regulated genes.

**Figure supplement 2.** RNA sequencing (RNA-Seq) data validation by quantitative reverse transcription-polymerase chain reaction (qRT-PCR).

*2014*; *Martínez-Reyes and Chandel, 2020*). Our results show that the *ybr238cΔ* mutation triggers oxidative metabolism and stress protective machineries activation and, as a result increases CLS. Collectively, our findings reveal that *YBR238C* is a TORC1-regulated gene involved in mitochondrial function coupled with cellular aging. Therefore, we refer to *YBR238C* as AAG1: <u>A</u>ging-<u>A</u>ssociated <u>G</u>ene 1.

## *YBR238C* negatively regulates mitochondrial function via *HAP4*-dependent and -independent mechanisms

*HAP4* is a TF that controls the expression of mitochondrial electron transport chain components including OXPHOS genes (*Lin et al., 2002*). *HAP4* activity has been shown to increase lifespan by enhancing the mitochondrial respiration in cells (*Lin et al., 2002*). Intriguingly, *HAP4* is among the 13 TFs overrepresented among the upregulated DEGs in *ybr238cΔ* mutant and control mitochondrial genes (*Figure 3A*, *Figure 2—source data 1*). We validated the upregulation of *HAP4* gene expression in the *ybr238cΔ* mutant under both yeast backgrounds BY4743 and CEN.PK (*Figure 3B, C* ). Consistent with these findings, transcription profile of mitochondrial genes in the *ybr238cΔ* mutant is opposite to that of the *hap4Δ* mutant (*Figure 3D*). For example, ETC complexes I–V genes' expression is increased in *ybr238cΔ*, however, it is decreased by *HAP4* deletion (*Figure 3D*).

We found that deletion of *HAP4* moderately decreases CLS at the background of the *ybr238cΔ* mutant (*Figure 3E*). To confirm that the decrease in lifespan is through the *HAP4* pathway, we examined the expression of mitochondrial respiratory genes. We found that *HAP4* deletion significantly decrease the ETC complexes I–V genes' expression in *ybr238cΔ* mutant (*Figure 3D*). To note, *HAP4* deletion at the wild-type background decreases the cellular lifespan even more (*Figure 3E*), which is consistent with more dramatic reduction of ETC complexes I–V genes' expression (*Figure 3D*). Taken together these data suggest that *YBR238C* negatively regulates the *HAP4* activity. *HAP4*-upregulated, increased mitochondrial function contributes to the prolonged lifespan of *ybr238cΔ* cells.

*YBR238C* gene deletion rescues some loss of lifespan of *hap4Δ* cells (*Figure 3F*). The effect is especially pronounced until day 8 when wild-type cells are essentially 100% surviving. Yet, complete epistasis of phenotypes is not achieved. This observation parallels the above findings that *HAP4* deletion in *ybr238cΔ* mutant does not fully recover the mitochondrial ETC complexes I–V genes' expression and CLS at day 9 compared to the wild-type background (*Figure 3D, E*). Together, these results indicate that *YBR238C* affects cellular lifespan via *HAP4*-dependent and -independent mechanisms.

To confirm the existence of *HAP4*-independent mechanisms, we examined the lifespan and transcriptome under conditions of *YBR238C* overexpression (*YBR238C-OE*). As expected, *YBR238C-OE* decreases the expression of *HAP4*, of mitochondrial ETC complexes I–V genes (*Figure 3G*) as well as the CLS (*Figure 3H*). Strikingly, the lifespan of *YBR238C-OE* cells was shorter than for *hap4Δ* cells (*Figure 3H*). Thus, a *HAP4*-independent mechanism does exist through which *YBR238C* also affects cellular aging (*Figure 3I*).

## The *YBR238C* paralog *RMD9* deletion decreases the lifespan of cells

In yeast *S. cerevisiae*, *YBR238C* has a paralog *RMD9* that shares ~45% amino acid identity (*Nouet et al., 2007*). Given that *YBR238C* activity is tightly coupled with the lifespan, we investigated the role of its paralog *RMD9* in cellular aging. Surprisingly, we found that *RMD9* deletion has an effect opposite to *YBR238C* deletion and it shortens CLS (*Figure 4A–C*, *Figure 4—figure supplement 1A–C*).

*RMD9* was initially isolated during a search for genes required for meiotic nuclear division (*Enyenihi and Saunders, 2003*). Subsequently, its role was uncovered in controlling the expression of mitochondrial metabolism genes by stabilizing and processing mitochondrial mRNAs (*Hillen et al., 2021*; *Nouet et al., 2007*; *Williams et al., 2007*). We asked whether *RMD9* deletion induces mitochondrial dysfunction that, therefore, causes accelerated cellular aging. We first tested the mitochondrial activity by allowing mutant cells to grow under respiratory conditions (*Hughes et al., 2020*; *Zulkifli et al., 2020*; *Merz and Westermann, 2009*). We found that *rmd9Δ* mutants could perform on glucose

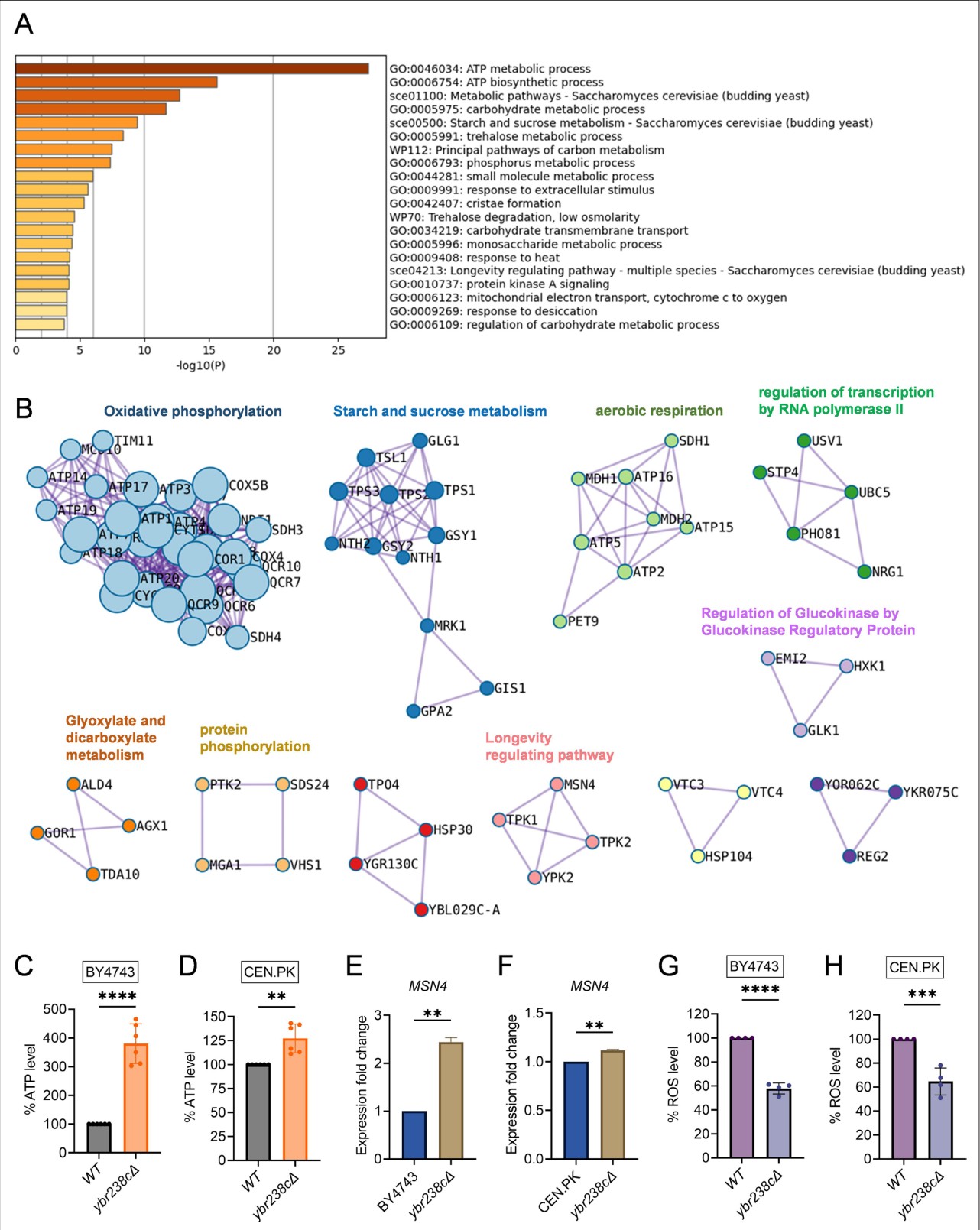

**Figure 2.** Longevity signatures of *ybr238cΔ* mutant. (**A, B**) Functional enrichment analysis of upregulated genes in yeast *Saccharomyces cerevisiae ybr238cΔ* mutant compared to wild type (BY4743). Bar plots showing the enriched biological process (**A**) and identified MCODE complexes based on the top 3 ontology enriched terms (**B**). Full set of genes for functional enrichment analysis (**A**) and the MCODE complexes (**B**) are available in *Figure 2—source data 1*. (**C and D**) Adenosine triphosphate (ATP) analysis of wild type and *ybr238cΔ* mutant for BY4743 (**C**) and CEN.PK113-7D (**D**). Quantification

*Figure 2 continued on next page*

*Figure 2 continued*

was performed using ATP extracted from 72-hr stationary cultures grown in synthetic defined medium supplemented with auxotrophic amino acids for BY4743 and only the synthetic defined medium for CEN.PK113-7D strains. Data are represented as means ± standard deviation (SD) ($n = 6$). **$p < 0.01$ and ****$p < 0.0001$ based on two-sided Student's $t$-test. (**E and F**) Expression analysis of *MSN4* gene by quantitative reverse transcription-polymerase chain reaction (qRT-PCR) in yeast *Saccharomyces cerevisiae* genetic backgrounds BY4743 (**E**) and CEN.PK113-7D (**F**). qRT-PCR was performed using RNA extracted from logarithmic-phase cultures grown in synthetic defined medium supplemented with auxotrophic amino acids for BY4743 and only the synthetic defined medium for CEN.PK113-7D strains. Data are represented as means ± SD ($n = 2$). **$p < 0.01$ based on two-sided Student's $t$-test. (**G and H**) Reactive oxygen species (ROS) analysis of wild type and *ybr238cΔ* mutant for BY4743 (**G**) and CEN.PK113-7D (**H**). Quantification was performed for logarithmic-phase cultures grown in synthetic defined medium supplemented with auxotrophic amino acids for BY4743 and only the synthetic defined medium for CEN.PK113-7D strains. Data are represented as means ± SD ($n = 4$). ***$p < 0.001$ and ****$p < 0.0001$ based on two-sided Student's $t$-test.

The online version of this article includes the following source data and figure supplement(s) for figure 2:

**Source data 1.** Transcriptomics and transcription factor analysis of *ybr238cΔ* cells.

**Figure supplement 1.** Identification of the role of uncharacterized *YBR238C* gene in cellular aging.

**Figure supplement 2.** Transcription factors (TFs) analysis for upregulated differentially expressed genes (DEGs) of *ybr238cΔ* mutant.

medium, but they failed to grow on glycerol as a carbon source (known to require functional mitochondria; *Figure 4D*, *Figure 4—figure supplement 1D*). Our results are in line with previously observed dysfunctional mitochondria in *rmd9Δ* mutants (*Hillen et al., 2021*; *Nouet et al., 2007*).

In control experiments, we found a similar growth defect phenotype on glycerol medium for cells deficient in *PET100* and *COX6* mitochondrial genes (*Figure 4D*, *Figure 4—figure supplement 1D*; *Merz and Westermann, 2009*). CLS of *pet100Δ* and *cox6Δ* mutants were reduced compared to wild type (*Figure 4E–G*, *Figure 4—figure supplement 1E–G*). We also found lowered ATP and higher ROS levels in *rmd9Δ*, *pet100Δ*, *cox6Δ* mutants compared to wild-type cells (*Figure 4H*, *Figure 4—figure supplement 1H*).

In contrast, long-lived *ybr238cΔ* mutants efficiently grow on respiratory medium with high ATP and low ROS levels (*Figure 4D, H*, *Figure 4—figure supplement 1D, H*). So far, the results indicate that deletion of *YBR238C* potentiates the mitochondrial function that, in turn, leads to CLS increase. Thus, *YBR238C* and its paralog *RMD9* antagonistically affect mitochondrial function, CLS, and cellular aging.

## *YBR238C* affects the cellular lifespan via an *RMD9*-dependent mechanism

Previously, *YBR238C* deletion was shown to increase the CLS via *HAP4*-dependent and -independent mechanisms (*Figure 3*) and to rescue some loss of CLS of the *hap4Δ* mutant by a *HAP4*-independent mechanism (*Figure 3E, F*). Here, we ask whether *YBR238C* deletion suppresses the shortened lifespan of *rmd9Δ* mutants. We examined the lifespan of double deletion *rmd9Δ ybr238cΔ* mutant and compared it with *rmd9Δ*. We found that the deletion of *YBR238C* largely failed to recover the lifespan of *rmd9Δ* mutant to wild-type cells; however, it partially prevented their early cell death (*Figure 5A–C*). These results show that *YBR238C* deletion increases the cellular lifespan via *RMD9*-dependent mechanisms. Also, we found that the *YBR238C* deletion results in increased ATP and reduced ROS levels in the *rmd9Δ* mutant (*Figure 5D, E*).

Intriguingly, we find *RMD9* expression upregulated in *ybr238cΔ* and downregulated in *YBR238C-OE* cells, respectively (*Figure 5F*). So, we asked whether *RMD9* expression change is transcriptionally coupled in cellular lifespan phenotypes. Remarkably, *RMD9* overexpression increases the lifespan of cells (*Figure 5G–I*) as we would have predicted from the observed changes with the *YBR238C* deletion phenotype.

To know whether this CLS increase is contributed by enhanced mitochondrial function, we quantified the ATP level in wild-type, *YBR238C-OE*, and *RMD9-OE* cells. ATP level is higher in *RMD9-OE* cells than wild-type cells, a result in line with *RMD9* positively regulating mitochondrial activity (*Figure 5J*). Consistent with the above findings, *YBR238C* overexpression decreases the ATP level (*Figure 5J*). Next, we assessed the oxidative stress and found that the ROS level in *RMD9-OE* cells was comparable to wild type. Yet, *YBR238C* overexpression increases the ROS level (*Figure 5K*).

It can be shown that the effects of *YBR238C* and *RMD9* are at least partially realized via the mitochondrial ETC pathway. Antimycin A (AMA) is an inhibitor of the ETC complex III, decreasing ATP synthesis (*Figure 6—figure supplement 1A*; *Huang et al., 2005*). Also, AMA treatment reduces the

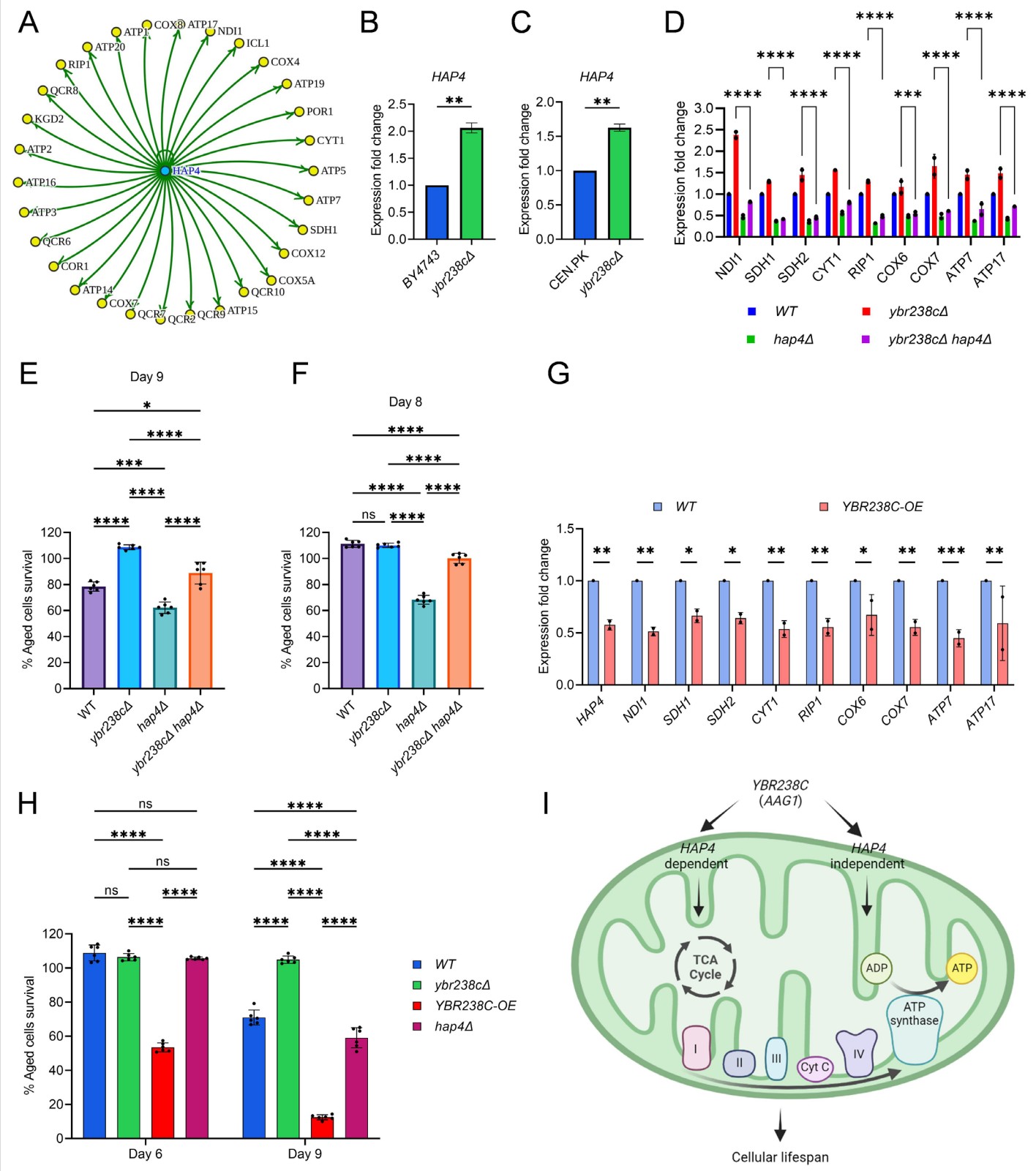

**Figure 3.** *YBR238C* affects cellular aging via *HAP4*-dependent and -independent mechanisms. (**A**) The *HAP4* regulon within the upregulated differentially expressed genes (DEGs) of *ybr238cΔ* mutant. See also *Figure 2—source data 1*. Expression analysis of *HAP4* gene by quantitative reverse transcription-polymerase chain reaction (qRT-PCR) in yeast *Saccharomyces cerevisiae* genetic backgrounds BY4743 (**B**) and CEN.PK113-7D (**C**). qRT-PCR was performed using RNA extracted from logarithmic-phase cultures grown in synthetic defined medium supplemented with auxotrophic amino acids

*Figure 3 continued on next page*

*Figure 3 continued*

for BY4743 and only the synthetic defined medium for CEN.PK113-7D strains. Data are represented as means ± standard deviation (SD) (*n* = 2). **p < 0.01 based on two-sided Student's *t*-test. (**D**) Expression analysis of mitochondrial ETC genes by qRT-PCR in wild type, *ybr238cΔ*, *hap4Δ*, and *ybr238cΔ hap4Δ*, strains of yeast *S. cerevisiae* genetic background CEN.PK113-7D. qRT-PCR was performed using RNA extracted from logarithmic-phase cultures grown in synthetic defined medium. Data are represented as means ± SD (*n* = 2). Comparing data between *ybr238cΔ* and *ybr238cΔ hap4Δ*. ***p < 0.001 and ****p < 0.0001 based on two-way analysis of variance (ANOVA) followed by Šídák's multiple comparisons test. (**E, F, H**) Chronological lifespan (CLS) of yeast *S. cerevisiae* genetic background CEN.PK113-7D strains was performed in synthetic defined medium using 96-well plate. Aged cells survival was measured relative to the outgrowth of day 2. Data are represented as means ± SD (*n* = 6). *p < 0.05, ***p < 0.001, ****p < 0.0001, and ns: non-significant. Ordinary one-way ANOVA followed by Tukey's multiple comparisons test (**E, F**). Two-way ANOVA followed by Šídák's multiple comparisons test (**H**). (**G**) Expression analysis of *HAP4* and ETC genes by qRT-PCR in wild-type and *YBR238C* overexpression (*YBR238C-OE*) strains of yeast *S. cerevisiae* genetic background CEN.PK113-7D (**C**). qRT-PCR was performed using RNA extracted from logarithmic-phase cultures grown in synthetic defined medium. Data are represented as means ± SD (*n* = 2). *p < 0.05, **p < 0.01, and ***p < 0.001 based on two-way ANOVA followed by Šídák's multiple comparisons test. (**I**) Model representing regulation of cellular aging by *YBR238C* via *HAP4*-dependent and -independent mechanisms.

lifespan of cells (*Figure 6—figure supplement 1B*), confirming that mitochondrial energy supply is critical to delay cellular aging. Since *YBR238C* and *RMD9* affect the ATP level, we tested the AMA effect on their deletion and overexpression strains. We found that *ybr238cΔ* and *RMD9-OE* cells with their enhanced mitochondrial function phenotype were largely resistant to AMA treatment (*Figure 5L*). AMA aggravates the cellular aging of mitochondrial defective *YBR238C-OE* cells (*Figure 5L*). Notably, mitochondrial defective *rmd9Δ* cells were not further affected by AMA treatment (*Figure 5L*).

Altogether, our findings reveal that *YBR238C* affects CLS and cellular aging via modulating mitochondrial function by mechanisms including *HAP4*- and *RMD9*-dependent pathways (*Figure 5M*).

## *YBR238C* connects TORC1 signaling with modulating mitochondrial function and cellular aging

So far, we learned that *YBR238C* is regulated by TORC1 (*Figure 1A, B*) and it affects cellular lifespan by modulating mitochondrial function. Here, we wish establish the direct connection between the TORC1 signaling and mitochondrial activity. We treated cells with the TORC1 inhibitor rapamycin and found that, subsequently, the ATP-level increases in the cells (*Figure 6A*, *Figure 6—figure supplement 1C*). To test whether TORC1 affects mitochondrial function via *YBR238C*, we analyze the expression profile of mitochondrial ETC genes. As expected, rapamycin supplementation decreased the expression of *YBR238C* (*Figure 6—figure supplement 1D, E*) but induced the expression of ETC genes (*Figure 6B*). Remarkably, rapamycin-induced changes in the expression of ETC genes were largely unaffected in *ybr238cΔ* cells and reduced in *YBR238C-OE* cells (*Figure 6B*). Our results are consistent with the hypothesis that TORC1 regulates the mitochondrial ETC genes via *YBR238C*.

The strains *ybr238cΔ* and *YBR238C-OE* are associated with increased and decreased CLS, respectively. So, we ask whether TORC1 influences cellular aging via *YBR238C* by modulating mitochondrial function. We examined the effect of rapamycin supplementation on the lifespan of *ybr238cΔ* and *YBR238C-OE* cells. Strikingly, addition of rapamycin does not further increase CLS of *ybr238cΔ* cells (*Figure 6C*). Additionally, anti-aging effect of rapamycin is significantly reduced in *YBR238C-OE* cells (*Figure 6C*). These findings are consistent with the rapamycin effect on transcriptomics profiles of ETC genes in *ybr238cΔ* and *YBR238C-OE* cells (*Figure 6B*). Taken together, these results reveal that *YBR238C* is a downstream effector of TORC1 signaling, connecting mitochondrial function for the regulation of cellular aging.

## Mitochondrial dysfunction induces the TORC1 activity that causes accelerated cellular aging

We showed that *YBR238C-OE* cells are associated with mitochondrial dysfunction and shortened lifespan (*Figure 5*). Apparently, the accelerated cellular aging phenotype is primarily due to compromised cellular energy level that affects the lifespan. Genetic (*ybr238cΔ*) and chemical (rapamycin) mediated enhancement of ATP level increases the lifespan of wild-type cells validate this conclusion. Notably, the rapamycin supplementation significantly rescued the shortened lifespan of mitochondrial defective *YBR238C-OE* cells (*Figure 6D*, *Figure 6—figure supplement 1F*). This observation suggests that, as a reaction on sensing mitochondrial dysfunction, TORC1 is activated which, in turn, leads to accelerated cellular aging. Indeed, we found that cellular growth (a proxy

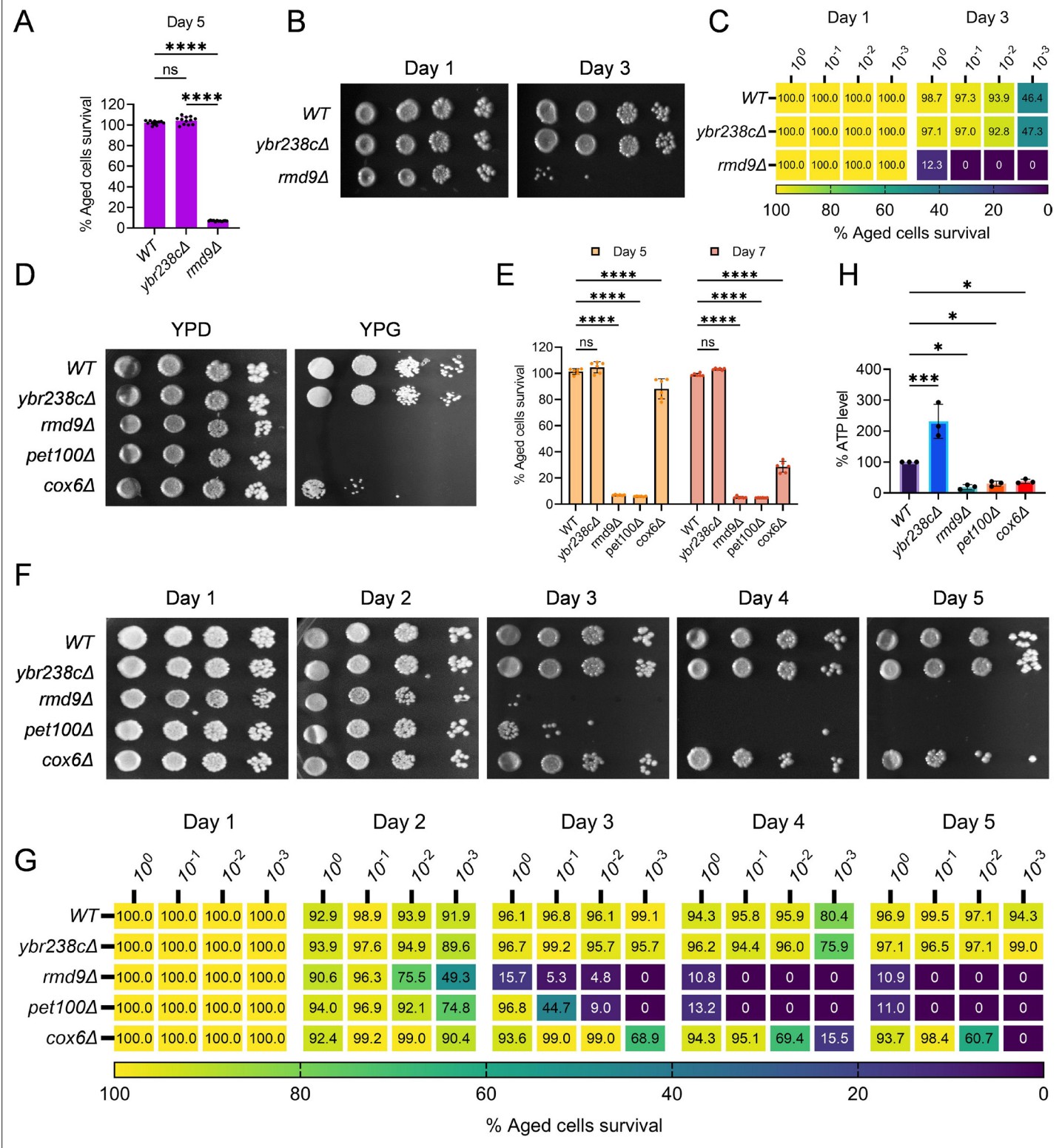

**Figure 4.** *RMD9* deletion decreases the cellular lifespan. (**A, E**) Chronological lifespan (CLS) of yeast *Saccharomyces cerevisiae* genetic background CEN.PK113-7D strains was performed in synthetic defined medium using 96-well plate. Aged cells survival was measured relative to the outgrowth of day 2. Data are represented as means ± standard deviation (SD) (*n* = 12) (**A**) and (*n* = 6) (**B**). ****p < 0.0001 and ns: non-significant. Ordinary one-way analysis of variance (ANOVA) followed by Tukey's multiple comparisons test (**A**). Two-way ANOVA followed by Dunnett's multiple comparisons test (**E**). (**B, C, F, G**) CLS of yeast strains was performed in synthetic defined medium using flask. Outgrowth was performed for 10-fold serial diluted aged cells onto the YPD agar plate (**B, F**) and YPD medium in the 96-well plate (**C, G**). The serial outgrowth of aged cells on agar medium was imaged (**B,**

*Figure 4 continued on next page*

Figure 4 continued

F) and quantified the survival relative to outgrowth of day 1 (**C, G**). A representative of two experiments for B, C, F, and G is shown. (**D**) Serial dilution growth assays of the yeast strains on fermentative glucose medium (YPD) and respiratory glycerol medium (YPG). (**H**) Adenosine triphosphate (ATP) level in yeast strains. Quantification was performed using ATP extracted from 72-hr stationary cultures grown in synthetic defined medium. Data are represented as means ± SD ($n$ = 3). *p < 0.05 and ***p < 0.001 based on ordinary one-way ANOVA followed by Dunnett's multiple comparisons test.

The online version of this article includes the following figure supplement(s) for figure 4:

**Figure supplement 1.** *YBR238C* homolog *RMD9* deletion leads to mitochondrial dysfunction associated with accelerated cellular aging.

for TORC1 activity) of mitochondria dysfunctional *YBR238C-OE* cells was resistant to rapamycin inhibition at concentrations that were effective for wild-type cells (*Figure 6—figure supplement 1G*).

Furthermore, we asked what the effect of TORC1 activity under enhanced mitochondrial function conditions would be. To test this, we assessed the growth of *ybr238cΔ* cells with rapamycin. We found that the cellular growth of *ybr238cΔ* cells was reduced by rapamycin compared to wild-type cells (*Figure 6—figure supplement 1G*). Apparently, TORC1 activity is not signaled downstream under these conditions and rapamycin further aggravates this by TORC1 inhibition. This result suggests that *YBR238C* affects TORC1 activity via modulating mitochondrial function.

We found a similar pattern of rapamycin effect on growth of mutant cells with enhanced or damaged mitochondrial function. Growth of *RMD9-OE* cells having enhanced mitochondrial function was sensitive to rapamycin (*Figure 6—figure supplement 1G*). Likewise, growth of defective mitochondrial *rmd9Δ*, *pet100Δ*, and *cox6Δ* mutants is resistant to rapamycin compared to wild type (*Figure 6—figure supplement 1G*). The *YBR238C* deletion reduced the rapamycin growth resistance phenotype of mitochondrial dysfunctional *rmd9Δ* (*Figure 6—figure supplement 1G*) and *hap4Δ* cells (*Figure 6—figure supplement 1H*), apparently by enabling enhanced mitochondrial function.

Taken together, we see that *YBR238C* plays a junction role in integrating mitochondrial function and TORC1 signaling.

## TORC1 inhibition prevents accelerated cellular aging caused by mitochondrial dysfunction in yeast and human cells

Mitochondrial dysfunction increases the TORC1 activity and, subsequently, causes accelerating cellular aging. We asked whether inhibition of TORC1 activity could prevent accelerating cellular aging even under mitochondrial dysfunction conditions. Previously, we have already shown that rapamycin-mediated TORC1 inhibition partly rescued the shortened lifespan of mitochondrial defective *YBR238C-OE* cells (*Figure 6D*, *Figure 6—figure supplement 1F*). We examined other mitochondrial dysfunctional conditions to confirm that the suppressive effect of rapamycin is not only specific to *YBR238C-OE*. We tested defective mitochondrial mutants *rmd9Δ*, *pet100Δ*, and *cox6Δ*. In all cases, rapamycin supplementation prevents the accelerated cellular aging (*Figure 6E–G*).

Rapamycin supplementation also rescued the AMA-mediated accelerated cellular aging (*Figure 6H*). We also quantified the ATP level to confirm that the suppressive effect of rapamycin is specific to mitochondrial function. Rapamycin supplementation increases the ATP level of AMA-treated cells (*Figure 6I*, *Figure 6—figure supplement 1I*), demonstrating that TORC1 inhibition improves mitochondrial function and thus suppresses accelerated cellular aging.

We further verified the connection of TORC1 and mitochondrial function in human cells. First, we examined the effect of AMA on the survival of HEK293 cells. AMA treatment decreases the viability of HEK293 cells (*Figure 6J*). Cell survival reduction is due to the mitochondrial dysfunction as we see lower ATP levels in AMA-treated cells compared to untreated control (*Figure 6—figure supplement 1J*). Subsequently, we examined the survival of AMA-treated HEK293 cells with or without rapamycin supplementation. Rapamycin supplementation suppresses AMA-mediated cellular death (*Figure 6K*). Further rapamycin supplementation increases the ATP-level-treated cells (*Figure 6—figure supplement 1J*).

Our findings convincingly support that TORC1 inhibition suppresses cellular aging associated with mitochondrial dysfunction across species.

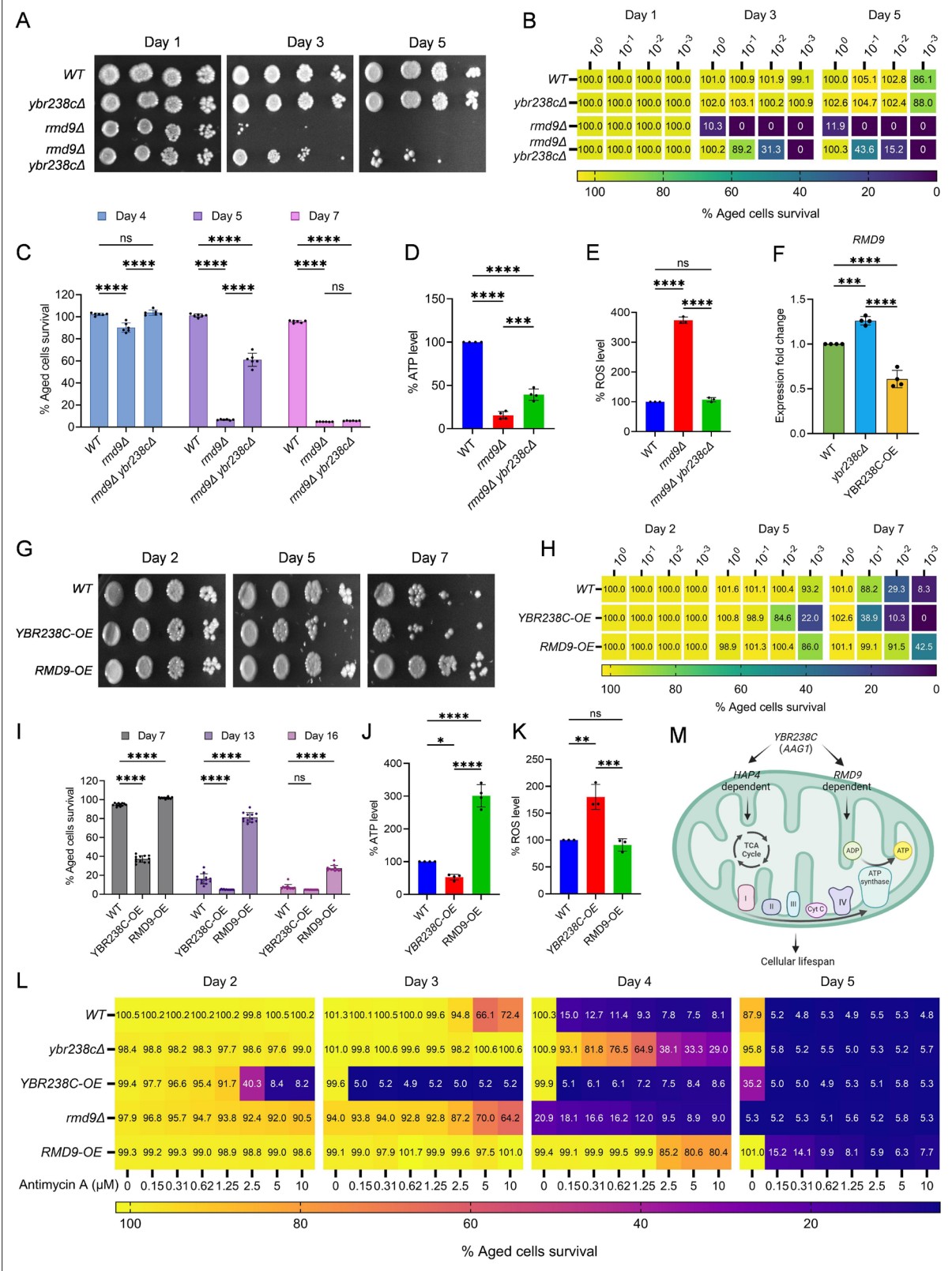

**Figure 5.** *YBR238C* affects cellular aging via *RMD9*-dependent mechanism. (**A, B, G, H**) Chronological lifespan (CLS) of yeast *Saccharomyces cerevisiae* genetic background CEN.PK113-7D strains was performed in synthetic defined medium using flask. Outgrowth was performed for 10-fold serial diluted aged cells onto the YPD agar plate (**A, G**) and YPD medium in the 96-well plate (**B, H**). The serial outgrowth of aged cells on agar medium was imaged (**A, G**) and quantified the survival relative to outgrowth of day 2 (**B, H**). A representative of two experiments for A, B, G, and H is shown. (**C, I**) CLS of

*Figure 5 continued on next page*

*Figure 5 continued*

yeast strains was performed in synthetic defined medium using 96-well plate. Aged cells survival was measured relative to the outgrowth of day 2. Data are represented as means ± standard deviation (SD) (*n* = 6) (**C**) and (*n* = 12) (**I**). ****p < 0.0001 based on two-way analysis of variance (ANOVA) followed by Tukey's multiple comparisons test (**C**) and Dunnett's multiple comparisons test (**I**). ns: non-significant. (**D, J**) Adenosine triphosphate (ATP) level in yeast strains. Quantification was performed using ATP extracted from 72-hr stationary cultures grown in synthetic defined medium. Data are represented as means ± SD (*n* = 4). *p < 0.05, ***p < 0.001, and ****p < 0.0001 based on ordinary one-way ANOVA followed by Tukey's multiple comparisons test. (**E, K**) Reactive oxygen species (ROS) level in yeast strains. Quantification was performed for logarithmic-phase cultures grown in synthetic defined medium. Data are represented as means ± SD (*n* = 3). **p < 0.01, ***p < 0.001, and ****p < 0.0001 based on ordinary one-way ANOVA followed by Tukey's multiple comparisons test. ns: non-significant. (**F**) Expression analysis of *RMD9* gene by quantitative reverse transcription-polymerase chain reaction (qRT-PCR) in yeast strains. qRT-PCR was performed using RNA extracted from logarithmic-phase cultures grown in synthetic defined medium. Data are represented as means ± SD (*n* = 4). ***p < 0.001 and ****p < 0.0001 based on ordinary one-way ANOVA followed by Tukey's multiple comparisons test. (**L**) Effect of antimycin A (AMA) treatment on CLS of yeast strain was assessed in synthetic defined medium using 96-well plate. Aged cells survival was measured relative to the outgrowth of day 1. A representative heatmap data of three experiments is shown. (**M**) Model representing regulation of cellular aging by *YBR238C* via *HAP4-* and *RMD9*-dependent mechanisms.

## Discussion

Identifying lifespan regulators, genetic networks, and connecting genetic nodes relevant to cellular aging provides functional insight into the complexity of aging biology *Kennedy et al., 2014*; *Cohen et al., 2022*; *López-Otín et al., 2013* for the subsequent design of anti-aging interventions. Understanding the mechanism of aging will also require understanding the role of many genes of yet unknown function as *YBR238C* at the beginning of this work. Unfortunately, the group of uncharacterized genes coding proteins of unknown function has received dramatically decreasing attention during the past two decades (*Eisenhaber, 2012*; *Sinha et al., 2018*; *Tantoso et al., 2023*).

In this study, we first summarized literature and genome database reports about genes affecting aging in the yeast model (mostly observed in gene deletion screens). We found that AAGs make up about one-third of the total yeast genome (many of them are functionally uncharacterized) and can be classified into 15 categories for increase/decrease CLS and/or RLS under various conditions.

We compared the set of AAGs with the group of RRGs (rapamycin-regulated genes) and found that about one-third of the AAGs is TORC1 activity regulated. Among the functionally uncharacterized genes in this overlap set, the gene *YBR238C* stands out as the only one that (1) is downregulated by rapamycin and (2) its deletion increases both CLS (*Burtner et al., 2011*) and RLS (*McCormick et al., 2015*; *Delaney et al., 2011*; *Kaeberlein et al., 2005*; *Schleit et al., 2013*). Thus, *YBR238C* is important among identified rapamycin-downregulated uncharacterized genes in AAG categories as it seems to rationally connect the rapamycin-induced TORC1 inhibition with the increase of both CLS and RLS.

We observed that the CLS of *ybr238cΔ* cells is higher than for wild type, regardless of various aging experimental conditions tested in this study. Transcriptional analysis of *ybr238cΔ* cells identified a longevity signature including enhanced gene expression related to mitochondrial energy metabolism and stress response genes (*Powers et al., 2006*; *Lin et al., 2002*; *Sun et al., 2016*; *Navarro and Boveris, 2007*; *Qiu et al., 2010*; *Bonawitz et al., 2007*). In contrast, *YBR238C-OE* cells display mitochondrial dysfunction leading to decreased lifespan. Together, these results reveal that TORC1 regulates the *YBR238C* expression which is linked to mitochondrial function and cellular lifespan.

The *YBR238C* paralog *RMD9* has been previously shown to affect mitochondrial function (*Hillen et al., 2021*; *Nouet et al., 2007*). Surprisingly, we observed an antagonism regarding the CLS phenotype of deletion/overexpression for *YBR238C* and *RMD9*. *YBR238C* overexpression and *RMD9* deletion confer defective mitochondrial function associated with accelerated cellular aging and decrease lifespan of cells. In contrast, *YBR238C* deletion and *RMD9* overexpression enhance the mitochondrial function that increase the cells' longevity. Consistent with the identified negative regulatory role of *YBR238C* on mitochondrial function, deletion of this gene partially suppressed the accelerated aging of *rmd9Δ* cells and AMA-treated cells. We identified that *YBR238C* influences mitochondrial function and the CLS phenotype with contributions from *HAP4-* and *RMD9*-dependent mechanisms. As (1) one of*RMD9*'s molecular functions has been shown to stabilize certain (mitochondrial) mRNAs and (2) *YBR238C* has a homologous pentatricopeptide repeat region, the two paralogs might differ in the sets of protected mRNAs with opposite outcome on cellular longevity.

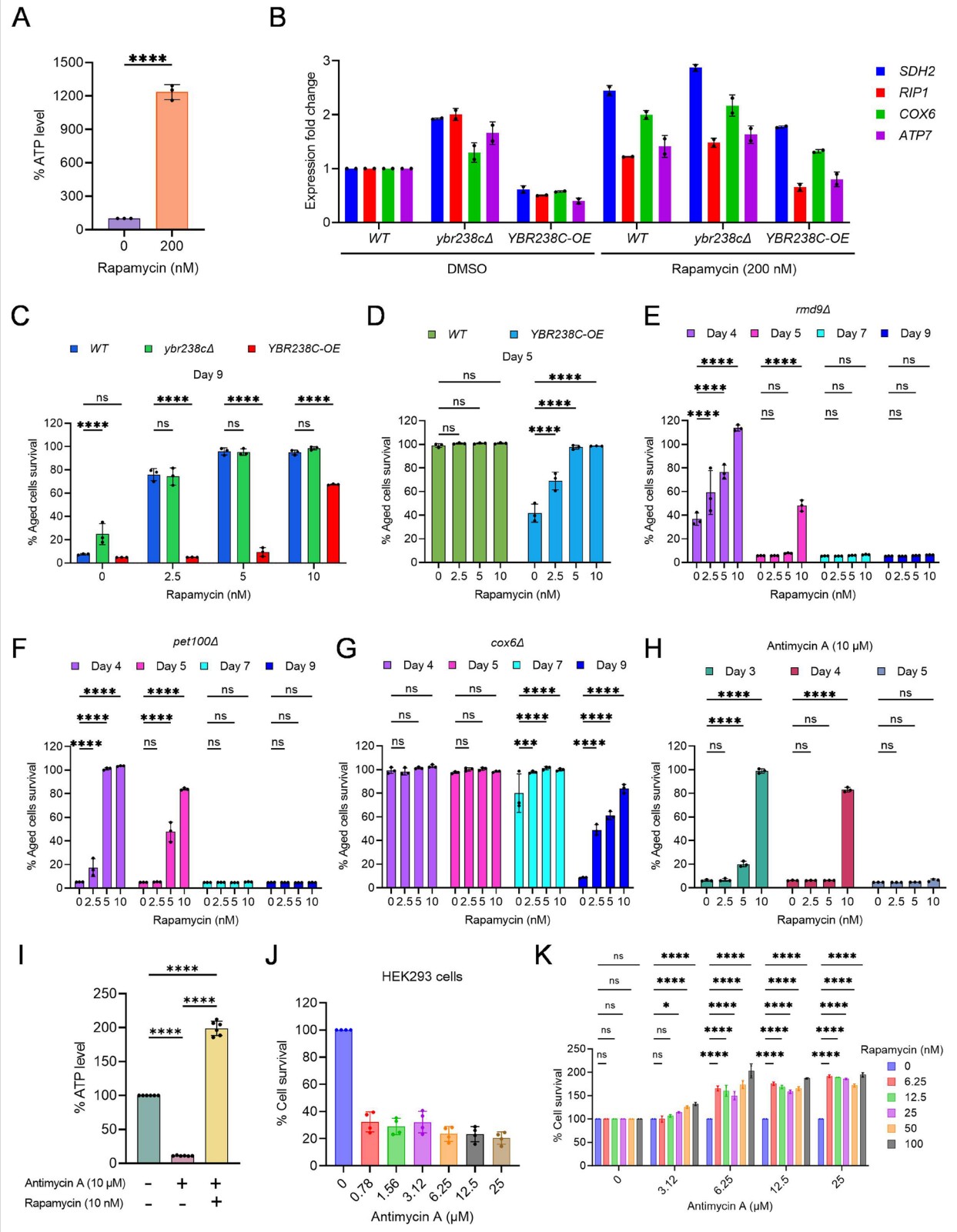

**Figure 6.** TORC1–MItochondria–TORC1 (TOMITO) signaling axis regulate cellular aging. (**A**) Adenosine triphosphate (ATP) analysis of logarithmic-phase wild-type CEN.PK113-7D yeast cells treated with rapamycin for 1 hr in synthetic defined medium. Data are represented as means ± standard deviation (SD) (*n* = 3). ****p < 0.0001 based on ordinary one-way analysis of variance (ANOVA) followed by Dunnett's multiple comparisons test. See also *Figure 6—figure supplement 1C*. (**B**) Expression analysis of mitochondrial ETC genes by quantitative reverse transcription-polymerase chain reaction

*Figure 6 continued on next page*

*Figure 6 continued*

(qRT-PCR). Analysis was performed using RNA extracted from logarithmic-phase wild-type CEN.PK113-7D yeast cell treated with rapamycin for 1 hr in synthetic defined medium. Data are represented as means ± SD (*n* = 2). (**C–G**) Chronological lifespan (CLS) of yeast strains with indicated concentrations of rapamycin was performed in synthetic defined medium using 96-well plate. Aged cells survival was measured relative to the outgrowth of day 2. Data are represented as means ± SD (*n* = 3). ***p < 0.001 and ****p < 0.0001 based on two-way ANOVA followed by Dunnett's multiple comparisons test. ns: non-significant. (**H**) CLS of wild-type yeast strain with indicated concentrations of rapamycin and antimycin A (10 μM) was performed in synthetic defined medium using 96-well plate. Aged cells survival was measured relative to the outgrowth of day 1. Data are represented as means ± SD (*n* = 3). ****p < 0.0001 based on two-way ANOVA followed by Dunnett's multiple comparisons test. ns: non-significant. (**I**) ATP analysis of stationary-phase wild-type CEN.PK113-7D yeast cells were incubated with rapamycin (10 nM) and antimycin A (10 μM) in synthetic defined medium. Data are represented as means ± SD (*n* = 6). ****p < 0.0001 based on ordinary one-way ANOVA followed by Tukey's multiple comparisons test. See also *Figure 6—figure supplement 1H*. (**J**) Determination of human HEK293 cells survival incubated with indicated concentrations of antimycin A for 6 days. Untreated control is considered 100% cell survival. Cells survival was measured relative to the control. Data are represented as means ± SD (*n* = 6). (**K**) Cell survival analysis of human HEK293 cells incubated with indicated concentrations of antimycin A and rapamycin for 6 days. Antimycin A and rapamycin individual treated concentrations considered 100% cell survival control. Cells survival of combined treated cells was measured relative to the control. Data are represented as means ± SD (*n* = 2). *p < 0.05,****p < 0.0001, and ns: non-significant based on two-way ANOVA followed by Dunnett's multiple comparisons test.

The online version of this article includes the following figure supplement(s) for figure 6:

**Figure supplement 1.** Target of Rapamycin Complex 1 (TORC1)–mitochondrial function signaling pathways control cellular aging.

While Kaeberlein et al. identified *YBR238C* as a top candidate for increasing RLS in yeast through deletion (*Kaeberlein et al., 2005*), our study builds upon their work by investigating the mechanisms and the connection with TORC1 in CLS. Results of genetic interventions (*YBR238C* deletion and over-expression) and rapamycin treatment show that *YBR238C* is a downstream effector by which TORC1 modulates mitochondrial activity. Most importantly, we found that TORC1 inhibition increases the mitochondrial function via *YBR238C* and, thereby, extends cellular lifespan. TORC1 is involved in both CLS and RLS (*Longo et al., 2012*). Whether the TORC1–*YBR238C* axis has similar or distinct mechanisms for CLS and RLS will be interesting to identify in the future.

TORC1 is the major controller of cellular metabolism that links environmental cues and signals for cellular growth and homeostasis. TORC1 upregulates anabolic processes such as de novo synthesis of proteins, nucleotides, lipids, and downregulates catabolic processes such as inhibition of autophagy (*Saxton and Sabatini, 2017*; *González and Hall, 2017*). We find that dysfunctional mitochondria lead to TORC1 activation both in yeast and in human cells with accompanying onset of accelerated cellular aging. Our results are in line with a recent report about anabolic pathways enhancement and suppression of catabolic processes in cells with defective mitochondria (*Hernberg and Nikkanen, 1970*). On the other hand, TORC1 activity decreases under enhanced mitochondrial function environment. The cause of the mitochondrial dysfunction (whether due to deletion of a critical gene or pharmacological intervention) is irrelevant in this context. Nevertheless, despite the mitochondrial insufficiency background, inhibition of TORC1 (e.g., with rapamycin) can partially rescue the shortened cellular lifespan (*Figure 6D–H*, *Figure 6—figure supplement 1F*).

Our work sheds light onto the questions (i) how mitochondrial dysfunction is linked to accelerated cellular aging and (ii) how TORC1 inhibition suppresses the mitochondrial dysfunction and prevents shortening lifespan. Apparently, mitochondrial dysfunction aberrantly signals to increase the TORC1 activity that leads to accelerated aging in cells. Remarkably, TORC1 inhibition can often suppress the accelerated cellular aging associated with impaired mitochondrial function. Yet, we found that growth of mutant strains with dysfunctional mitochondria can be resistant despite TORC1 inhibition by rapamycin with concentrations effective in wild-type cells possibly due to the dramatic increase of TORC1 activity as in *YBR238C-OE*, *rmd9Δ*, *pet100Δ*, *cox6Δ*, and *hap4Δ* cells (*Figure 6—figure supplement 1G, H*).

Our interpretation of the experimental results is supported by two recently published studies: (1) TORC1 activation was observed after mitochondrial ETC dysfunction in an induced pluripotent cell model (*Zheng et al., 2016*). (2) The inhibition of TORC1 delays the progression of brain pathology of mice with ETC complex I NDUFS4-subunit knockout (*Johnson et al., 2013b*).

We think that it will be insightful to explore TORC1 activity under various conditions with compromised mitochondrial function including mitochondrial fission and fusion dynamics which is reported to affect in cell viability (*Archer, 2013*; *Youle and van der Bliek, 2012*). Altogether, our findings uncover the central role of the feedback loop between mitochondrial function and TORC1 signaling (TORC1–MItochondria–TORC1 (TOMITO) signaling process). Whereas the effector from TORC1 to

mitochondria involves *YBR238C*, the other direction appears executed via metabolite sensing (e.g., α-keto-glutarate, glutamine, etc.) (*Durán et al., 2012*).

## Methods

### Data acquisition

The gene lists that modulate cellular lifespan in aging model organism yeast *S. cerevisiae* were extracted from database SGD (*Engel et al., 2022*) and GenAge (*Townes et al., 2020*) (as of November 8, 2022). The actual gene lists are available in *Figure 1—source data 1*.

### Yeast strains, growth media, and cell culture

The *S. cerevisiae* auxotrophic BY4743 (Euroscarf) and prototrophic CEN.PK113-7D (*Bauer et al., 2000*) strains were used in this study. Deletion strains for BY4743 obtained from yeast homozygous diploid collection. Deletion strains in CEN.PK background was generated using standard PCR-based method (*Longtine et al., 1998*). Yeast strains were revived from frozen glycerol stock on YPD agar (1% Bacto yeast extract, 2% Bacto peptone, 2% glucose, and 2.5% Bacto agar) medium for 2–3 days at 30°C. For all experiments, CEN.PK113-7D strains were grown in synthetic defined (SD) medium contain 6.7 g/l yeast nitrogen base with ammonium sulfate without amino acids and 2% glucose. SD medium supplemented with histidine (40 mg/l), leucine (160 mg/l), and uracil (40 mg/l) for auxotrophic BY4743 strains.

### Human cell lines, growth media, and cell culture

Human embryonic kidney (HEK293) cell line (American Type Culture Collection (ATCC), CRL-1573) was cultured in high-glucose Dulbecco's modified Eagle's medium supplemented with 10% fetal bovine serum and 1% penicillin–streptomycin solution. All cells cultured in a humidified incubator with 5% $CO_2$ at 37°C. The cell lines were authenticated by the suppliers using short tandem repeat profiling, ensuring their identity. Additionally, it was confirmed that the cells were not contaminated with mycoplasma. Stocks were subsequently prepared from these authenticated and contamination-free cells. For each new experiment, a fresh aliquot from these stocks was used to maintain consistency and integrity in our experimental procedures.

### Chemical treatment to cell culture

Stock solution of rapamycin and AMA was prepared in DMSO. The final concentration of DMSO did not exceed 1% in yeast and 0.01% in human cell lines experiments.

### Yeast aging assay

For the CLS experiments, prototrophic CEN.PK113-7D strains were grown in SD medium contain 6.7 g/l yeast nitrogen base with ammonium sulfate without amino acids and 2% glucose. SD medium supplemented with histidine (40 mg/l), leucine (160 mg/l), and uracil (40 mg/l) for auxotrophic BY4743 strains. Chronological aging was assessed by determining the lifespan of yeast as described previously with slight modifications (*Longo et al., 2012*; *Alfatah and Eisenhaber, 2023*). Yeast culture grown in overnight at 30°C with 220 rpm shaking in glass flask was diluted to starting optical density at 600 nm ($OD_{600}$) ~0.2 in fresh medium to initiate the CLS experiment. CLS was performed by outgrowth method utilizing three different approaches: (1) Cells grown and aged in 96-well plates with a total 200 µl culture in SD medium at 30°C. At various age time points yeast stationary culture (2 µl) were transferred to a second 96-well plate containing 200 µl YPD medium and incubated for 24 hr at 30°C without shaking. Outgrowth $OD_{600}$ for each age point was measured by the microplate reader. (2) Cells grown and aged in flask with a total culture volume more than 5 ml SD medium and incubated for 24 hr at 30°C with 220 rpm shaking. At different age time points yeast stationary culture washed and normalized to $OD_{600}$ of 1.0 with YPD medium. Furthermore, normalized yeast cells were serial 10-fold diluted with YPD medium in 96-well plates. 3 µl of diluted culture were spotted onto the YPD agar plate and incubated for 48 hr at 30°C. The outgrowth of aged cells on the YPD agar plate was photographed using the GelDoc imaging system. (3) The above discussed serial 10-fold diluted yeast stationary culture with YPD medium in 96-well plates incubated for 24 hr at 30°C without shaking. Outgrowth $OD_{600}$ for serial diluted aged cells was measured by the microplate reader.

## RNA extraction

Yeast cells were first mechanically lysed using the manufacturer's disruption protocol. Total RNA from yeast cells was extracted using QIAGEN RNeasy mini kit. ND-1000 UV-visible light spectrophotometer (Nanodrop Technologies) and Bioanalyzer 2100 with the RNA 6000 Nano Lab Chip kit (Agilent) was used to assess the concentration and integrity of RNA.

## RNA-Seq and bioinformatics analysis

RNA-Seq was conducted using NovaSeq PE150. Raw Fastq files were then passed into Fastp v0.23.2 for adapter trimming and low-quality reads removal. Both single- and paired-end reads were processed with default parameters with `--detect` adapter for pe. Raw reads that passed the quality check were then aligned using HiSat 2 v2.2.1 with the index built from *S. cerevisiae* R-64-1-1 top-level DNA fasta file obtained from Ensembl for sequencing data with S288C strain background. Library information for all experiments were first checked with RSeQC infer_experiment.py v4.0.0 before the addition of the respective parameters for alignment and counts. The resulting SAM files were then converted and sorted to BAM files using Samtools v1.13. The BAM files were then used to generate feature counts using HTSeq v1.99.2. HTSeq counts from each experiment were then used for downstream DEGs analysis. Counts generated from HTSeq were then used for differential gene expression analysis using EdgeR v3.34.1 quasi likelihood *F*-test and DESeq2 v1.36.0. Principal component analysis (PCA) was conducted via the use of transcript per million normalized counts. Samples that are separated by batches in the PCA were corrected using ComBat-Seq v3.44.0 before PCA was conducted after normalization of the corrected counts. Functional enrichment analysis was performed by metascape tool (*Bitto et al., 2016*).

## qRT-PCR analysis

qRT-PCR experiments were performed as described previously using QuantiTect Reverse Transcription Kit (QIAGEN) and SYBR Fast Universal qPCR Kit (Kapa Biosystems) (*Alfatah et al., 2021*). The abundance of each gene was determined relative to the house-keeping transcript *ACT1*.

## ATP analysis

ATP analysis was performed as described previously (*Alfatah et al., 2023*). Yeast cells were mixed with the final concentration of 5% trichloroacetic acid (TCA) and then kept on ice for at least 5 min. Cells were washed and resuspended in 10% TCA and lysis was performed with glass beads in a bead beater to extract the ATP. ATP extraction from human HEK293 cells was performed using Triton X-100 lysis buffer. The ATP level was quantified by PhosphoWorks Luminometric ATP Assay Kit (AAT Bioquest) and normalized by protein content measured Bio-Rad protein assay kit.

## mtDNA copy number analysis

Determination of mtDNA copy number was analyzed by real-time qPCR (quantitative polymerase chain reaction). DNA was extracted using Quick-DNA Midiprep Plus Kit (Zymo Research). qPCR was performed in a final volume of 20 µl containing 20 ng of total DNA using SYBR Fast Universal qPCR Kit (Kapa Biosystems) and analyzed using the Quant Studio 6 Flex system (Applied Biosystems). The real-time qPCR conditions were one hold at (95°C, 180 s), followed by 40 cycles of (95°C, 1 s) and (60°C, 20 s) steps. A melting-curve analysis was included in the cycle after amplification to verify PCR specificity and the absence of primer dimers. Relative mtDNA content was determined by qPCR of mitochondrial genes (*ATP6* and *COX3*) and normalized with nuclear-specific gene *ACT1*.

## ROS measurement

Cells were washed and resuspended in 1× phosphate buffer saline (PBS, pH 7.4). After that cells were incubated with 40 µM H2DCFDA (Molecular probe) for 30 min at 30°C (*Haque et al., 2019*). Cells were then washed with PBS and ROS level was measured by fluorescence reading (excitation at 485 nm, emission at 524 nm) by the microplate reader. The fluorescence intensity was normalized with $OD_{600}$.

## TFs analysis

TFs enrichment analysis was performed using YEASTRACT (*Teixeira et al., 2006*; *Teixeira et al., 2018*). The significant p-value <0.05 considered for regulatory network analysis based on DNA-binding plus expression evidence. The TFs regulatory networks were visualized with a force-directed layout.

## Fermentative and respiratory growth assay

Yeast cells grown in YPD medium were washed and normalized to $OD_{600}$ of 1.0 with water. Furthermore, normalized yeast cells were serial 10-fold diluted with water. 3 µl of diluted culture were spotted onto the agar medium YPD (2% glucose) and YPG (3% glycerol) and incubated for 48 hr at 30°C. The cell growth on the agar plate was photographed using the GelDoc imaging system.

## Growth sensitivity assay

Effect of chemical compounds on cell growth was carried out in 96-well plates. At an $OD_{600}$ of ~0.2 in SD medium 200 µl yeast cells was transferred into the 96-well plate containing serially double-diluted concentrations of compounds. Cells were incubated at 30°C and the growth was measured at $OD_{600}$ by the microplate reader.

## Quantification and statistical analysis

Data analysis of all the experimental results such as mean value, standard deviations, significance, and graphing were performed using GraphPad Prism v.9.3.1 software. The comparison of obtained results was statistically performed using the Student's *t*-tests, ordinary one-way analysis of variance (ANOVA), and two-way ANOVA followed by multiples comparison tests. In all the graph plots, p values are shown as *$p < 0.05$, **$p < 0.01$, ***$p < 0.001$, and ****$p < 0.0001$ were considered significant. ns: non-significant.

## Lead contact and materials availability

Further information and requests for resources and reagents should be directed to and will be fulfilled by the Lead Contact, Dr. Mohammad Alfatah (alfatahm@bii.a-star.edu.sg).

## Acknowledgements

We thank Sebastian Maurer-Stroh (Executive Director, BII, A*STAR), Lee Hwee Kuan (Training and Talent Deputy Director, BII, A*STAR), Chandra S Verma (Research Deputy Director, BII, A*STAR), Su Xinyi (Acting Executive Director, IMCB, A*STAR), Farid John Ghadessy (Senior Principal Scientist), and Cheok Chit Fang (Principal Investigator, IMCB, A*STAR) for backing this research. This work was supported by Bioinformatics Institute, A*STAR Career Development Fund (C210112008), US NAM Healthy Longevity Catalyst Awards Grant (MOH-000758-00), and YIRG, National Medical Research Council, Singapore (MOH-001348-00).

## Additional information

### Funding

| Funder | Grant reference number | Author |
|---|---|---|
| A*STAR Career Development Fund, Singapore | C210112008 | Mohammad Alfatah |
| US NAM Healthy Longevity Catalyst Awards Grant | MOH-000758-00 | Mohammad Alfatah Frank Eisenhaber |
| YIRG, National Medical Research Council, Singapore | MOH-001348-00 | Mohammad Alfatah |

The funders had no role in study design, data collection, and interpretation, or the decision to submit the work for publication.

### Author contributions

Mohammad Alfatah, Conceptualization, Supervision, Funding acquisition, Writing – original draft, Writing – review and editing; Jolyn Jia Jia Lim, Yizhong Zhang, Arshia Naaz, Trishia Yi Ning Cheng, Sonia Yogasundaram, Nashrul Afiq Faidzinn, Jovian Jing Lin, Birgit Eisenhaber, Formal analysis, Investigation; Frank Eisenhaber, Formal analysis, Investigation, Writing – review and editing

### Author ORCIDs

Mohammad Alfatah ⓘ https://orcid.org/0000-0003-1073-8933
Trishia Yi Ning Cheng ⓘ https://orcid.org/0009-0008-5735-1633

Reviewer #1 (Public Review): https://doi.org/10.7554/eLife.92178.3.sa1
Reviewer #2 (Public Review): https://doi.org/10.7554/eLife.92178.3.sa2
Reviewer #3 (Public Review): https://doi.org/10.7554/eLife.92178.3.sa3
Author response https://doi.org/10.7554/eLife.92178.3.sa4

## Additional files

### Supplementary files

• MDAR checklist

### Data availability

Sequencing data have been deposited in SRA under accession codes SAMN38984574, SAMN38984575, SAMN38984576, SAMN38984577.

The following datasets were generated:

| Author(s) | Year | Dataset title | Dataset URL | Database and Identifier |
|---|---|---|---|---|
| Alfatah M | 2023 | 2% Glucose_WT_1 | https://www.ncbi.nlm.nih.gov/biosample/SAMN38984574 | NCBI BioSample, SAMN38984574 |
| Alfatah M | 2023 | 2% Glucose_ΔYBR238C_1 | https://www.ncbi.nlm.nih.gov/biosample/SAMN38984575 | NCBI BioSample, SAMN38984575 |
| Alfatah M | 2023 | 2% Glucose_WT_2 | https://www.ncbi.nlm.nih.gov/biosample/SAMN38984576 | NCBI BioSample, SAMN38984576 |
| Alfatah M | 2023 | 2% Glucose_ΔYBR238C_2 | https://www.ncbi.nlm.nih.gov/biosample/SAMN38984577 | NCBI BioSample, SAMN38984577 |

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
