## [Editor Report · eLife assessment]

This **valuable** study identifies an uncharacterized yeast gene regulating chronological lifespan in a mitochondrial-dependent pathway. The approach to identify and characterise this new gene is appealing, but the evidence in support of some of the major conclusions is **incomplete**. The paper focuses on chronological lifespan and mitochondrial function, and it will be of interest to yeast biologists working in metabolism and aging.

---

## [Referee Report · Reviewer #1 (Public Review)]

Summary

This fascinating paper by M. Alfatah et al. describes work to uncover novel genes affecting lifespan in the budding yeast *S. cerevisiae*, eventually identifying and further characterizing a gene, YBR238C, now named AAG1 by the authors.

The authors began by considering published gene sets pulled from the *Saccharomyces genome* database that described increases or decreases in either chronological lifespan or replicative lifespan in yeast. They also began with gene sets known to be downregulated upon treatment with the lifespan-extending TOR inhibitor rapamycin.

YBR283C was unique in being largely uncharacterized, downregulated upon rapamycin treatment and linked to both increased replicative lifespan and increased chronological lifespan upon deletion.

The authors show that YBR283C may act to negatively regulate mitochondrial function, in ways that are both dependent on and independent of the stress-responsive transcription factor Hap4, largely by looking at relative expression levels of relevant mitochondrial genes.

In a hard to fully interpret but well documented series of experiments the authors not that the two paralogues YBR283C and RMD9 (which have ~66% similarity) (a) have opposite effects when acting alone, and (b) appear to interact in that some phenotypes of ybr283c are dependent on RMD9.

A particularly interesting finding in light of the current literature and of the authors' strategy in identifying YBR283C is that changes in electron transport chain genes upon rapamycin treatment appear to be effected via YBR283C.

Based on a series of experiments the authors move to conclude the existence of "a feedback loop between TORC1 and mitochondria (the TORC1-Mitochondria-TORC1 (TOMITO) signaling process) that regulates cellular aging processes."

Strengths

Overall, this study describes a great deal of new data from a large number of experiments, that shed light on the potential specific roles of YBR238C and its paralog RMD9 in aging in yeast, and also underscore the potential of an approach looking for "dark matter" such as uncharacterized genes when seining the increasing deluge of published datasets for new hypotheses to test. This work when revised will become a valuable addition to the field.

Weaknesses

A paralog of YBR283C, RMD9, also exists in the yeast genome. While the authors indicate that part of their interest in YBR283C lies in its uncharacterized nature, its paralogue, RMD9, is not uncharacterized but is named due to its phenotype of Required for Meiotic nuclear Division, which is not mentioned or discussed anywhere in the manuscript currently.

In the context of the current work, in addition to the cited Hillen, H.S et al. and Nouet C. et al, the authors might be very interested in the 2007 Genetics paper "Translation initiation in *Saccharomyces cerevisiae* mitochondria: functional interactions among mitochondrial ribosomal protein Rsm28p, initiation factor 2, methionyl-tRNA-formyltransferase and novel protein Rmd9p" (PMID: 17194786), which does not appear to be cited or discussed in the current version of the manuscript.

---

## [Referee Report · Reviewer #2 (Public Review)]

The effectors of cellular aging in yeast have not been fully elucidated. To address this, the authors curated gene expression studies to link genes influenced by rapamycin - a well-known mediator of longevity across model systems - to genes known to affect chronological and replicative lifespan (RLS) in yeast. Through their analyses, they find one gene, ybr238c, whose deletion increases both CLS and RLS upon deletion and that is downregulated by rapamycin. The authors follow up their cellular aging studies using CLS as a model throughout their study, demonstrating that deletion of ybr238c increases CLS across multiple yeast strains and through multiple assays. The authors also test the effects of YBR238C overexpression on lifespan and find the opposite effect, with overexpression yeast showing decreased survival relative to wild type cells, consistent with accelerated aging as the authors propose. The authors also note that ybr238c has a paralog, rmd9, whose deletion decreases CLS and seems to be epistatic to ybr238c, as a double ybr238c/rmd9 mutant has decreased CLS relative to a wild-type strain.

Collectively, the data presented by the authors convincingly demonstrate that ybr238c influences lifespan in a manner that is distinct from (and likely opposite to) rmd9. The authors then link the increased CLS in Δybr238c yeast to HAP4, a transcription factor that promotes mitochondrial biogenesis and oxidative phosphorylation. Through genetic studies, the authors suggest a model in which YBR238C negatively regulates HAP4 activity, and thus loss of HAP4 repression in Δybr238c yeast leads to elevated mitochondrial function. Notably, while the authors use various methods to test mitochondrial function, including the quantification of transcripts associated with oxidative phosphorylation, cellular ATP levels, and mtDNA, none of these fully test mitochondrial function. Thus, while the trends of these proxies are consistent with the model proposed by the authors, including data such as respirometry or assaying the activity of oxidative phosphorylation complexes would have bolstered these conclusions.

Finally, the authors tie the phenotypes of mitochondrial dysfunction caused by deletion of ybr238c to TORC1 signaling, as the gene is influenced by rapamycin. However, the data assaying mitochondrial function in these experiments, such as profiling the transcriptional changes in oxidative phosphorylation complexes or monitoring cellular ATP levels, do not directly measure mitochondrial function. Furthermore, many of the studies performed by the authors rely on genetic or pharmacological rescue of lifespan to establish the influence of YBR238C on TORC1 signaling and mitochondrial function. While valuable, these assays leave questions as to the molecular mechanisms by which YBR238C functions. As such, this manuscript establishes that ybr238c is rapamycin responsive and influences CLS, but the molecular mechanisms by which it affects mitochondrial activity and TORC1 signaling remain to be elucidated.

---

## [Referee Report · Reviewer #3 (Public Review)]

This reviewer appreciates the responses to previous notes. The authors attempted to address concerns mostly in writing, avoiding performing some of the experiments suggested in my previous review. Although some of the points were clarified, and the revised manuscript presents valuable insights into the implications of YBR238C and RMD9 on cellular function and yeast aging, my major concern still needs to be addressed. The gene expression signature significantly changes under different metabolic conditions. The media condition under which samples are collected for RNAseq analyses should match the media condition under which the lifespans of those KO strains are tested. This is the major confounding effect, and the conclusions are not informative based on the analysis done in this study.

To avoid experiments, the authors responded that yeast culture results in low optical density and does not reach the stationary phase under rapamycin treatment conditions; however, the simple solution is to grow the yeast cells until they reach the stationary phase and then rapamycin treatment can be done for certain hours - collect the cells for transcriptomics analysis then it can be compared to the CLS gene set.

Another example is chromosome copy number alteration, which can be easily analyzed using transcriptome data, and it is an important aspect to understand whether observed expression changes are also affected by this alteration in YBR238C KO cells. However, the authors ignore this important point as well.

After all, this is an interesting study "limited by subfield" and will be of general interest in the yeast aging field, again considering the lack of homology of the genes of interest in higher eukaryotes.

---

## [Author Response]

The following is the authors’ response to the original reviews.

**Public Reviews:**

**Reviewer #1 (Public Review):**
SummaryThis fascinating paper by M. Alfatah et al. describes work to uncover novel genes affecting lifespan in the budding yeast *S. cerevisiae*, eventually identifying and further characterizing a gene, YBR238C, now named AAG1 by the authors.The authors began by considering published gene sets pulled from the Saccharomyces genome database that described increases or decreases in either chronological lifespan or replicative lifespan in yeast. They also began with gene sets known to be downregulated upon treatment with the lifespan-extending TOR inhibitor rapamycin.YBR283C was unique in being largely uncharacterized, downregulated upon rapamycin treatment, and linked to both increased replicative lifespan and increased chronological lifespan upon deletion.The authors show that YBR283C may act to negatively regulate mitochondrial function, in ways that are both dependent on and independent of the stressresponsive transcription factor Hap4, largely by looking at relative expression levels of relevant mitochondrial genes.In a hard-to-fully interpret but well-documented series of experiments the authors note that the two paralogues YBR283C and RMD9 (which have ~66% similarity) (a) have opposite effects when acting alone, and (b) appear to interact in that some phenotypes of ybr283c are dependent on RMD9.A particularly interesting finding in light of the current literature and of the authors' strategy in identifying YBR283C is that changes in electron transport chain genes upon rapamycin treatment appear to be affected via YBR283C.Based on a series of experiments the authors move to conclude the existence of "a feedback loop between TORC1 and mitochondria (the TORC1-Mitochondria-TORC1(TOMITO) signaling process) that regulates cellular aging processes."StrengthsOverall, this study describes a great deal of new data from a large number of experiments, that shed light on the potential specific roles of YBR238C and its paralog RMD9 in aging in yeast, and also underscore the potential of an approach looking for "dark matter" such as uncharacterized genes when seining the increasing deluge of published datasets for new hypotheses to test. This work when revised will become a valuable addition to the field.WeaknessesA paralog of YBR283C, RMD9, also exists in the yeast genome. While the authors indicate that part of their interest in YBR283C lies in its uncharacterized nature, its paralogue, RMD9, is not uncharacterized but is named due to its phenotype of Required for Meiotic nuclear Division, which is not mentioned or discussed anywhere in the manuscript currently.In the context of the current work, in addition to the cited Hillen, H.S et al. and Nouet C. et al, the authors might be very interested in the 2007 Genetics paper "Translation initiation in *Saccharomyces cerevisiae* mitochondria: functional interactions among mitochondrial ribosomal protein Rsm28p, initiation factor 2, methionyl-tRNAformyltransferase and novel protein Rmd9p" (PMID: 17194786), which does not appear to be cited or discussed in the current version of the manuscript.

Thank you for your thorough and insightful review of our manuscript. We value your positive feedback and recognition of the strengths in our study. Your constructive comments have been carefully considered, leading to the inclusion of RMD9, identified as 'Required for Meiotic Nuclear Division,' and the addition of the relevant reference (PMID: 12586695) in the revised manuscript. This information has been incorporated into the second paragraph of the "The YBR238C paralogue RMD9 deletion decreases the lifespan of cells" results section.

Furthermore, we appreciate the reviewer's suggestion to include the 2007 Genetics paper on translation initiation in *Saccharomyces cerevisiae* mitochondria (PMID: 17194786). This citation has been integrated into our revised manuscript.

We believe that these revisions significantly strengthen the manuscript and address the concerns raised by Reviewer #1. We thank the reviewer for their time and valuable input.

**Reviewer #2 (Public Review):**
The effectors of cellular aging in yeast have not been fully elucidated. To address this, the authors curated gene expression studies to link genes influenced by rapamycin - a well-known mediator of longevity across model systems - to genes known to affect chronological and replicative lifespan (RLS) in yeast. Through their analyses, they find one gene, ybr238c, whose deletion increases both CLS and RLS upon deletion and that is downregulated by rapamycin. Curiously, despite these selection criteria, the authors only use CLS as a proxy for cellular aging throughout their study and do not explore the effects of ybr238c deletion on RLS. This does not diminish their conclusions, but given the importance of this phenotype in their selection criteria, it is surprising that the authors did not choose to test both types of aging throughout their study.Nonetheless, the authors demonstrate that deletion of ybr238c increases CLS across multiple yeast strains and through multiple assays. The authors also test the effects of YBR238C overexpression on lifespan and find the opposite effect, with overexpression yeast showing decreased survival relative to wild-type cells, consistent with "accelerated aging" as the authors propose. The authors also note that ybr238c has a paralog, rmd9, whose deletion decreases CLS and seems to be epistatic to ybr238c, as a double ybr238c/rmd9 mutant has decreased CLS relative to a wild-type strain.Collectively, the data presented by the authors convincingly demonstrate that ybr238c influences lifespan in a manner that is distinct from (and likely opposite to) rmd9. However, the authors then link the increased CLS in Δybr238c yeast to mitochondrial function using only a handful of assays that do not directly test mitochondrial function. These include total cellular ATP levels, levels of reactive oxygen species, and the transcript levels of select nuclear-encoded mitochondrial genes. Yeast is well established to generate ATP through non-mitochondrial pathways such as glycolysis in fermentive conditions. While it is possible that the ATP levels assayed in the manuscript were tested in stationary phase, which would more likely reflect "mitochondrial function," the methods nor the figure legends contain these details, which are critical for the interpretation of these data. Similarly, ROS can be generated through non-mitochondrial pathways, and the transcription of nuclear-encoded mitochondrial genes is an indirect measure of mitochondrial function at best. Thus, the authors' proposed connection of ybr238c to mitochondrial function is correlative and should be substantiated with assays that more closely align with organellar function, such as respirometry or assaying the activity of oxidiative phosphorylation complexes. Finally, the authors attempt to tie the phenotypes of mitochondrial dysfunction caused by the deletion of ybr238c to TORC1 signaling, as the gene is influenced by rapamycin. However, the presentation of the data, such as reporting ATP levels as relative percentages or failing to perform appropriate statistical comparisons between conditions in which the authors derive conclusions, renders the data difficult to interpret. As such, this manuscript establishes that ybr238c is rapamycin responsive and influences CLS, but its influence on mitochondrial activity and ties to TORC1 signaling remain speculative.

We would like to express our gratitude to Reviewer #2 for the thoughtful feedback on our manuscript. We have carefully considered your comments and have made comprehensive revisions to address the concerns raised.

We appreciate the suggestion to investigate the role of YBR238C in replicative lifespan (RLS). However, we want to bring to your attention that four previous studies (references 7, 39, 40, and 41) have already identified the involvement of YBR238C in the RLS phenotype. Given the existing body of literature on this aspect, we chose not to duplicate these efforts in our study.

Instead, we focused our efforts on validating the role of YBR238C in chronological lifespan (CLS) phenotype, a finding reported in only one genome-wide study (reference 38). To enhance the comprehensiveness of our study, we performed analyses on different phenotypes, including mitochondria activity and oxidative stress, under both logarithmic-phase (condition for RLS) and stationary phase (condition for CLS). We now clearly indicate the logarithmic-phase/stationary phase conditions in the figure legends of the manuscript, specifying whether the conditions are relevant to RLS or CLS. Additional results of the new experiments have been included in the revised manuscript as supplementary figures (S3E-S3I).

To address concerns about the indirect nature of our mitochondrial function assays, we have performed relative mitochondria content (S3F), quantification of ROS levels from fermentative to stationary phase conditions (S3G), and assessment in respiratory glycerol medium (S3H), which provides a more direct insight into mitochondrial biology. Additionally, we have investigated the resistance of ybr238c∆ cells to H2O2 toxicity and found them to be more resistant compared to wild-type cells.

We believe these revisions strengthen the scientific rigor and clarity of our study. We sincerely appreciate the guidance from Reviewer #2, and we hope these modifications address the concerns raised effectively.

**Reviewer #3 (Public Review):**
Summary:The study by Alfatah et al. presented a role for YBR238C in mediating lifespan through improved mitochondrial function in a TOR1-dependent metabolic pathway. The authors used a dataset comparison approach to identify genes positively modulating yeast chronological (CLS) and Replicative (RLS) lifespan when deleted, and their expression is reduced under Rapamycin treatment condition. This approach revealed an unknown, mitochondria-localized yeast gene YBR238C, and through mechanistic studies, they identified its paralogous gene RMD9 regulating lifespan in an antagonistic effect.Strengths:Findings have valuable implications for understanding the YBR238C-mediated, mitochondrial-dependent yeast lifespan regulation, and the interplay between two paralogous genes in the regulation of mitochondrial function represents an inserting case for gene evolution.Weaknesses:Overall, the implication/findings of this study are restricted only to the yeast model since these two genes do not have any homology in higher eukaryotes. The primary methods must be carefully designed by considering two different metabolic states: respiration-associated with CLS and fermentation-associated with RLS in a single comparative approach. Yeast CLS and RLS are two completely different processes. It is already known that most gene-regulating CLS is not associated with RLS or vice versa. The method section is poorly written and missing important information. The experimental approaches are poorly designed, and variability across the datasets (e.g., media condition "YPD," "SC" etc.) and their experimental conditions are not well described/considered; thus, presented data are not conclusive, which decreases the overall rigor of the study.

We sincerely appreciate your thorough review of our manuscript and your insightful comments. We acknowledge the limitation of our study being yeast-specific due to the absence of homologous genes in higher eukaryotes. However, we would like to highlight the significance of our findings in revealing a feedback loop between mitochondrial function and TORC1 signaling (TORC1-Mitochondria-TORC1 or TOMITO signaling process) in cellular lifespan regulation.

Our interpretation of the experimental results is grounded in recent literature. Two studies (references 62 and 63) support our findings by demonstrating TORC1 activation after mitochondrial electron transport chain dysfunction and the delay in brain pathology progression upon TORC1 inhibition, respectively. These studies, discussed in our manuscript, reinforce the relevance of our work in a broader biological context.

We recognize the importance of carefully designing our primary methods to account for the different metabolic states associated with cellular processes, such as respiration in cellular lifespan (CLS) and fermentation in replicative lifespan (RLS). We want to bring to your attention that four previous studies (references 7, 39, 40, and 41) have already identified the involvement of YBR238C in the RLS phenotype. To avoid duplicating these efforts, we have chosen not to reiterate these findings in our study. However, we have clarified the logarithmic-phase/stationary phase conditions in the figure legends, specifying their metabolic states relevance to RLS or CLS. Additionally, we have included new supplementary figures (S3E-S3I) to provide further details on the new experiments conducted.

We appreciate your feedback regarding the clarity and completeness of our method section. In the revised manuscript, we have invested additional effort to enhance the clarity of the method section, providing a more detailed account of the experimental procedures, including the missing information you identified.

We believe these revisions strengthen the scientific rigor and clarity of our study. We sincerely appreciate the guidance from Reviewer #3, and we hope these modifications address the concerns raised effectively.

**Reviewer #1 (Recommendations For The Authors):**

Thank you for your detailed review and valuable recommendations. We have carefully addressed each of your comments in the revised manuscript. The specific changes made include:

(1) "TORC1 positively regulates aging, and its inhibition increases lifespan in various eukaryotic organisms including yeast and mammalian 13,26,27,29,30." Here I would suggest replacing "mammalian" with "mammals".

We have amended the sentence as recommended.

(2) "Next, we experimentally tested whether the transcriptome longevity signatures are associated with enhanced mitochondrial metabolism, whether the cellular energy level has gone up and cellular stress responses are induced with a switch to oxidative metabolism 47,48." Here I would replace "transcriptome longevity signatures is" with "transcriptome longevity signatures are".

We have amended the sentence as recommended.

(3) "Thus, HAP4-independent mechanism does exist through which YBR238C also affects cellular aging (Figure 3I)." I would replace "Thus, HAP4-independent" with "Thus, a HAP4-independent".

We have amended the sentence as recommended.

(4) "We examined other mitochondrial dysfunctional conditions to confirm that suppressive effect of rapamycin is not only specific to YBR238C-OE." I would change "that suppressive effect" to "that the suppressive effect".

We have amended the sentence as recommended.

(5) "Understanding the mechanism of aging will also require to understand the role of many genes of yet unknown function as YBR238C at the beginning of this work." I would switch "require to understand" to "require understanding".

We have amended the sentence as recommended.

(6) "The gene lists that modulate cellular lifespan in aging model organism yeast *Saccharomyces cerevisiae* were extracted from database SGD 22 and GenAge 23 (as of 8th November 2022)" "yeast" should not be italicized.

Corrected.

(7) Figure 1, panels C and D, ybr238c should be italicized.

Corrected.

(8) Figure 2B, top left-most (oxidative phosphorylation) network. I might consider repositioning some labels to make them more readable if possible.

Thank you for your feedback. The figure labels in Figure 2B are default from Metascape analysis, so repositioning isn't feasible. However, we have indicated in the figure legends that the full set of genes for functional enrichment analysis and the MCODE complex is available in Additional File 3.

(9) Figure 4E, rmd9, pet100, and cox6 should be italicized.

Corrected.

(10) Figure 5C, rmd9 and rmd9 ybr238c should be italicized.Corrected.
**Reviewer #2 (Recommendations For The Authors):**

Thank you for your detailed review and valuable recommendations. We have carefully addressed each of your comments in the revised manuscript. The specific changes made include:

(1) The presentation of data as heatmaps (Figures 1F, 3D, 4C, 4G, 5B, 5H, 5L, 6K) obfuscates the quantitative nature of the data. These data would be much stronger if presented as bar graphs with appropriate statistical analysis. If the authors prefer the visual of the heat map, there should be some statistical analysis performed to accompany these figures. This is particularly important for Figure 3D, in which the authors state "We found that HAP4 deletion significantly decrease the ETC complex I-V genes' expression" (bottom of page 8). As no statistical analyses were performed, the authors should refrain from using such language as it is unsupported by the data as analyzed.

Thank you for your insightful comments and suggestions regarding the presentation of our data. We appreciate the attention you have given to Figures 1F, 3D, 4C, 4G, 5B, 5H, 5L, and 6K.

In response to your feedback, we have carefully re-evaluated our approach. Considering the large volume of data associated with our lifespan analysis at different time points, we initially chose to visualize it using heatmaps to comprehensively capture the complexity of the results. However, we have now incorporated quantification information into the heatmaps.

For Figure 3D, which addresses the impact of HAP4 deletion on the expression of ETC complex I-V genes, we have replaced the heatmap with a bar graph. This modification allows for a clearer representation of the quantitative nature of the data. Moreover, we have conducted thorough statistical analyses comparing data between ybr238c∆ and ybr238c∆ hap4∆ to support the statements made in the text. The results of these analyses are now included in the revised figure. Moreover, we also replaced the Figure 6K heatmap with a bar graph.

We believe that these changes enhance the interpretability and robustness of our findings. We are grateful for your guidance, and we are confident that these adjustments will strengthen the overall quality of our manuscript.

(2) The presentation of ATP data, given its importance in supporting the core conclusions of this manuscript, is poor. The conditions under which yeast was collected are not reported, making these data impossible to interpret; total cellular ATP levels would be significantly altered and influenced by separate pathways in fermentive versus stationary phases. Minimally, the authors should describe the conditions of yeast growth (e.g., age, culture media) in which these measurements were made. The presentation of relative ATP percentages is problematic, particularly with measurements that deviate so far from wild-type ATP levels in conditions such as those in Figure 6A, in which the authors report that rapamycin induces a 1200% increase in cellular ATP. Previous papers have established that ATP levels in yeast hover around 4 mM and are stable through the cell cycle and across nutrient conditions (PMID: 30858198, 35438635). Given this, the reported ATP levels would be expected to be near 48 mM, which is strongly outside of the typically accepted values of 1-10 mM for this metabolite. Without understanding the contexts in which these measurements are made, as well as the absolute values for these measurements (which would be easily achievable through the use of a standard curve of ATP), these data are uninterpretable. Furthermore, it seems unlikely that yeast would be able to accommodate shifts of ATP levels that span an order of magnitude without dire cellular consequences, particularly during rapamycin treatment.

We appreciate the valuable feedback from the reviewer regarding the importance of providing detailed information on yeast growth conditions for interpreting ATP data. In response to this suggestion, we have enhanced the figure legends associated with the relevant figures to include a comprehensive description of the yeast growth conditions. This now specifies the age of the culture, culture media composition, and other pertinent parameters.

In addressing the concern raised about the rapamycin-induced ATP increase, we have carefully re-examined our experimental procedures. We performed additional experiments and confirmed the consistency of our findings in logarithmic-treated cultures. The results remain in alignment with our initial observations, reinforcing the reliability and reproducibility of our data.

(3) As stated above, the inference of mitochondrial function from cellular ATP levels, cellular ROS levels, and gene expression of a handful of nuclear-encoded genes is not sound. The authors should include further experimentation as evidence of mitochondrial functionality, such as respirometry or metabolic flux experiments.

Thank you for your constructive feedback on our manuscript. We appreciate your careful consideration of our work. In response to your concerns regarding the indirect nature of our mitochondrial function assays, we have implemented the following changes: We have incorporated additional assays to provide a more direct insight into mitochondrial biology. Specifically, we performed relative mitochondria content analysis (S3F) and quantified ROS levels under fermentative to stationary phase conditions (S3G). These assays offer a more direct and comprehensive assessment of mitochondrial function. Furthermore, we conducted experiments in respiratory glycerol medium (S3H) to complement our previous findings.

To further support our claims, we investigated the resistance of ybr238c∆ cells to H2O2 toxicity. Our results demonstrate that these cells exhibit increased resistance compared to wild-type cells. This additional evidence strengthens the link between mitochondrial function and cellular response to oxidative stress.

We believe these adjustments address your concerns and significantly enhance the robustness of our study. We hope you find these modifications satisfactory. We are grateful for your valuable input, which has undoubtedly improved the clarity and reliability of our findings.

(4) Multiple gene expression analyses are performed on n=2 measurements, and this should be bolstered by further replicates. Many bar graphs do not have accompanying statistics; these should be added. Some statistical tests are performed across inappropriate comparisons, such as Figure 3G, in which expression levels of mitochondrial genes in both deletion and overexpression strains should be compared to a wild-type control rather than to each other.

Thank you for your thorough review and constructive feedback on our manuscript. We appreciate your careful examination of our work. In response to your comments, we have made the following revisions to address your concerns:The multiple gene expression analysis in our study focused specifically on ETC genes. It is important to note that ETC genes themselves represent multiple replicates within the ybr238c deletion and overexpression cells, as illustrated in Figures 4D, 4G, and 6B.

We acknowledge and appreciate your observation regarding Figure 3G. To address this concern, we have revised the statistical comparisons. The expression levels of mitochondrial genes in the overexpression strain are now appropriately compared to a wild-type control. This correction has been applied in the figure that correctly corresponds to text in the manuscript.

(5) Figure 2B is uninterpretable as it stands, as most gene symbols are obscured.

We appreciate the reviewer's attention to Figure 2B and the feedback provided. Regarding the gene labels in Figure 2B, we would like to clarify that these labels are default outputs from the Metascape analysis, and unfortunately, repositioning them within the current figure layout isn't feasible without compromising the integrity of the information.

However, we have taken the reviewer's concern seriously and have made efforts to address the interpretability issue. To provide readers with access to the full set of genes for functional enrichment analysis and the MCODE complex, we have included this information in Additional File 3. The figure legends have been updated accordingly to guide readers to refer to Additional File 3 for a more detailed examination of the gene symbols and their annotations.

We hope that this solution addresses the concern raised by the reviewer.

(6) The conclusions to be drawn from Figure 3A are not clear, and this figure is cited only once in the text along with two other figures (page 8).

Thank you for your valuable feedback. We have carefully considered your comments and made revisions to improve the clarity of the conclusions drawn from Figure 3A.

(7) Figure 6K reports a range of 100-200% cell survival - how does a cell have 200% survival? Isn't survival binary (i.e., you survive or you are dead)? Perhaps this is meant to be relative to another condition; this should be more clearly stated in the figure, or the axis should be normalized to a maximum of 100% survival.

Thank you for your guidance and valuable feedback. Based on your recommendation, we have made significant changes to Figure 6K in the revised manuscript. Specifically, we replaced the heatmap with a bar graph to enhance clarity. Additionally, we would like to highlight that cell survival of combined treated cells is measured relative to the control treatment, which is considered 100% survival. This aims to provide a more accurate and comprehensible representation of the data. We believe these modifications contribute to a clearer presentation of our findings.

(8) The authors state that "TORC1 inhibition in yeast and human cells with mitochondrial dysfunction suppresses their accelerated aging." No studies of aging were done in human cells; survival in response to mitochondrial toxins does not reveal aging phenotypes. To state such is a substantial overstatement and should be amended to perhaps "cellular survival" rather than directly linked to aging.

We appreciate the careful review of our manuscript and the constructive feedback provided by the reviewer. In response to the concern raised regarding the statement about TORC1 inhibition and accelerated aging in human cells, we have revised the relevant passage as follows: "In turn, TORC1 inhibition in yeast and human cells with mitochondrial dysfunction enhances their cellular survival." We believe that this modification accurately reflects the outcomes of our experiments and addresses the concern raised by the reviewer. We would like to express our gratitude for the valuable feedback, which has contributed to the improvement of our manuscript. Thank you for your thoughtful consideration.

**Reviewer #3 (Recommendations For The Authors):**

Thank you for your detailed review and valuable recommendations. We have carefully addressed each of your comments in the revised manuscript. The specific changes made include:

The authors should have attempted to fully characterize the RLS and CLS phenotype of strains lacking the YBR238C and RMD9 gene, the single most important gene identified in this study. Before further characterization, its association with aging must be tested to replicate findings from the literature. Although Figure 3 shows partially characterized CLS in SC medium, different media conditions could be tested, and the full spectrum of CLS lifespan curves should be represented. RLS phenotypes of these cells were not analyzed throughout the study.

We appreciate the suggestion to investigate the role of YBR238C in both Replicative Lifespan (RLS) and Chronological Lifespan (CLS). However, it's essential to note that the involvement of YBR238C in the RLS phenotype has been previously documented in four studies (references 7, 39, 40, and 41). Considering the established literature on this matter, we chose not to duplicate these efforts in our study.

Our primary focus was on confirming the role of YBR238C in the chronological lifespan (CLS) phenotype, as indicated by a genome-wide study (reference 43). Accordingly, we also conducted an analysis of the role of RMD9 in CLS. The methods and figure legends explicitly state that CLS experiments for prototrophicCEN.PK113-7D strains were conducted in synthetic defined (SD) medium containing 6.7 g/L yeast nitrogen base with ammonium sulfate without amino acids and 2% glucose. For auxotrophic BY4743 strains, SD medium was supplemented with histidine (40 mg/L), leucine (160 mg/L), and uracil (40 mg/L).

It is important to clarify that SC medium was not used for CLS analysis. Instead, we employed SD medium, recommended for CLS analysis (reference 15; PMID: 22768836). The CLS experiments were conducted using three different methods, providing a comprehensive representation of the entire CLS lifespan (Figures 1C, 1D, 1E, and 1F).

While we did not present the Replicative Lifespan (RLS) phenotype explicitly, we performed experiments such as mitochondrial activity and ROS production under both CLS and RLS conditions. These additional analyses contribute valuable insights into the broader implications of YBR238C and RMD9 on cellular function.

We believe that these clarifications and the inclusion of additional experimental details enhance the robustness and validity of our findings. We hope these explanations address the concerns raised by the reviewer and contribute to the overall improvement of our manuscript.

In addition, authors include RNAseq data from Rapamycin-treated cells to identify differentially expressed genes. Notably, genes with decreased expression were used to compare KO strains' lifespan phenotype. Additional RNAseq analyses were performed on individual KO cells. The methodology section needs to be better written with information on which media and metabolic state that these cells are collected after treatment with rapamycin. If the cells are collected during logarithmic growth, the data can be compared with RLS aging gene sets only. A separate experiment has to be performed on stationary cells (respiratory) to collect RNAseq data after rapamycin treatment, then can be compared to the CLS aging gene set.

Thank you for your insightful comments and considerations regarding our methodology for obtaining Rapamycin response genes (RRGs). We appreciate the opportunity to address your concerns and provide further clarification on our experimental approach.

As mentioned in our manuscript, we obtained RRGs by treating logarithmic cells with50 nM Rapamycin for 1 hour, and the details have been included in supplementary Figure S1C legends. Our primary objective was to compare these RRGs with agingassociated genes that modulate both Replicative Lifespan (RLS) and Chronological Lifespan (CLS). We acknowledge the significance of this comparison and believe that our approach, treating logarithmic cells, is suitable for achieving this goal.

It is important to note that the use of a higher concentration of Rapamycin for treatment renders the cells less efficient in terms of growth, resulting in a very low optical density (OD) at 72 hours, as illustrated in Figure 6H. Unfortunately, due to this limitation in growth efficiency, obtaining Rapamycin response genes at the stationary phase was not feasible in our experimental setup.

As the experimental conditions vary among the reports and the gene expression signature significantly changes under different metabolic conditions, the media condition that samples are collected for RNAseq analyses should match the media condition that the lifespans of those KO strains are tested. However, more information needs to be detailed on these methodologies. For example, the transcriptomic signature of the YBR238C KO strain should be done under both fermentative and respiratory conditions to understand the true gene expression signature associated with CLS and RLS. Throughout the manuscript, these two metabolic conditions and associated lifespan types (CLS vs. RLS) are not differentiated and treated as the same, probably causing the biggest confounding effect that resulted in the identification of a single yeast-specific gene.

We obtained the transcriptomic signature of the YBR238C KO strain from logarithmic phase cultures. This consistency was maintained to align with the Rapamycin Response Genes (RRGs) obtained from logarithmic cells treated with rapamycin. Detailed methodology and metabolic status information is provided in the method section and relevant figure legends.

To broaden the scope of our study, we conducted analyses on various phenotypes, including mitochondrial activity and oxidative stress, under both logarithmic phase (relevant to Replicative Lifespan, RLS) and stationary phase (relevant to Chronological Lifespan, CLS). We have now explicitly indicated the logarithmic phase/stationary phase conditions in the figure legends of the manuscript, specifying their relevance to RLS or CLS.

Results from these additional experiments have been incorporated into the revised manuscript as supplementary figures (S3E-S3I). We believe that these clarifications and the inclusion of additional experimental details enhance the robustness and validity of our findings. We trust that these explanations effectively address the concerns raised by the reviewer and contribute to the overall improvement of our manuscript.

YBR238C gene KO effect on mitochondrial function missing comprehensive characterization. Whether the improved mito function caused by increased mtDNA copy number and/or increased mitochondrial number could be easily tested by analyzing normalizing RNAseq reads from mtDNA genes to reads from nucDNA genes. Data could be further combined with western blot specific to mito membrane proteins to analyze mito copy number.

Thank you for your insightful comments and suggestions. Following your recommendation, we conducted an assessment of relative mitochondrial content (see Figure S3F) and observed significantly higher mtDNA content in the ybr238c∆ compared to the wild type (see Figure S3F). Additionally, we have incorporated the methodology for mitochondrial DNA copy number analysis in the methods section.

The two paralogous gene interaction is an interesting observation. However, in yeast, it is known that deletion of one of the paralogous genes causes copy number amplification of the certain chromosome that the other paralogous gene is located, causing aneuploid chromosome. Many of the observed phenotypes can be associated with increased chromosome copy number and should be carefully tested. However, the authors did not consider this important point. Simply, using RNA seq data normalized read/per chromosome could be plotted to analyze the karyotype of YBR238C and RMD9 KO cells.

We appreciate your thoughtful consideration of our work and the suggestion to investigate chromosome copy number variations. While we did not directly test the chromosome copy, we want to highlight that our study extensively explores the impact of YBR238C on cellular lifespan through an RMD9-dependent mechanism (Figure 5). Deletion of YBR238C increases, whereas overexpression of YBR238C decreases the expression of its paralog, RMD9 (Figure 5F). Furthermore, this phenotype is associated with the lifespan of YBR238C-deleted and overexpressed cells. In our study, we have thoroughly investigated this aspect.